# A design principle of polymers processable into 2D homeotropic order

Zhen Chen[1,2], Yi-Tsu Chan[1,3], Daigo Miyajima[2,4], Takashi Kajitani[3,5], Atsuko Kosaka[3,5], Takanori Fukushima[3,5], Jose M. Lobez[1,3] & Takuzo Aida[1,2,3]

How to orient polymers homeotropically in thin films has been a long-standing issue in polymer science because polymers intrinsically prefer to lie down. Here we provide a design principle for polymers that are processable into a 2D homeotropic order. The key to this achievement was a recognition that cylindrical polymers can be designed to possess oppositely directed local dipoles in their cross-section, which possibly force polymers to tightly connect bilaterally, affording a 2D rectangular assembly. With a physical assistance of the surface grooves on Teflon sheets that sandwich polymer samples, homeotropic ordering is likely nucleated and gradually propagates upon hot-pressing towards the interior of the film. Consequently, the 2D rectangular lattice is constructed such that its b axis (side chains) aligns along the surface grooves, while its c axis (polymer backbone) aligns homeotropically on a Teflon sheet. This finding paves the way to molecularly engineered 2D polymers with anomalous functions.

[1] Department of Chemistry and Biotechnology, School of Engineering, The University of Tokyo, 7-3-1 Hongo, Bunkyo-ku, Tokyo 113-8656, Japan. [2] RIKEN Center for Emergent Matter Science, 2-1 Hirosawa, Wako, Saitama 351-0198, Japan. [3] RIKEN Advanced Science Institute, 2-1 Hirosawa, Wako, Saitama 351-0198, Japan. [4] RIKEN SPring-8 Center, 1-1-1 Kouto, Sayo, Hyogo 679-5148, Japan. [5] Chemical Resources Laboratory, Tokyo Institute of Technology, 4259 Nagatsuta, Midori-ku, Yokohama 226-8503, Japan. Correspondence and requests for materials should be addressed to T.K. (email: kajitani@res.titech.ac.jp) or to T.F. (email: fukushima@res.titech.ac.jp) or to T.A. (email: aida@macro.t.u-tokyo.ac.jp).

Polymer-based devices are extensively used for practical applications, where the performances of such devices are highly dependent on the physical properties of constituent polymeric materials. Because of this, chemists have been motivated to design new architectures of polymers having superb features[1–5]. However, in order to fully exploit the potential utilities of such polymeric materials, the ability to control their structural order and molecular packing is equally important[6–15]. In fact, much attention has been focused on this essential issue. However, studies so far made are mostly concentrated on how polymer chains can rationally be oriented unidirectionally on the horizontal planes of substrates (Fig. 1a,i)[13–17], whereas little attention has been paid to the realization of vertical (homeotropic) orientation of polymers. This is because of a preconceived notion that polymer chains intrinsically prefer to lie flat on substrates. However, as in the case of separation membranes[18,19], transistors[20] and solar cells[21–23], for example, some emerging applications of polymers allowed us to recognize the presence of a strong demand for two-dimensional (2D) materials consisting of homeotropically oriented polymer chains, ideally over a large area (Fig. 1a,ii).

Some conjugated polymers have been reported to align homeotropically on substrates to form thin films for organic photovoltaics[24–28]. However, such oriented films are as thick as only 300 nm at most[25]. Furthermore, non-conjugated and flexible polymer chains have barely been explored except those that are dendritic[29,30] or grown from substrates by the polymerization with surface-immobilized initiators[31–34]. In 2010, we reported a freestanding photoactuator film consisting of bottlebrush polymer **PMA^{AAA}** with three azobenzene units in its individual side chains[35]. This film was obtained by hot-pressing **PMA^{AAA}** between two Teflon sheets such that their surface grooves were oriented parallel to one another. Astonishing findings were that the polymer chains are homeotropically oriented to the film plane and that this anomalous orientation is essential for the film to undergo photoactuation. What are the underlying principle and required structural parameters for this anomalous assembling behaviour of **PMA^{AAA}**? In the present work, we tackled this grand challenge in polymer science and, as a result of our effort to conduct a thorough systematic study with 13 newly designed polymers with triple-mesogenic long side chains, we successfully established a design principle of polymers that are processable into a 2D homeotropic order. The key to this important achievement was a recognition that cylindrical polymers can be designed to possess oppositely oriented local dipoles in their cross-section. The interaction of such local dipoles possibly forces cylindrical polymers to tightly connect bilaterally, affording a 2D rectangular assembly. With a physical assistance of the surface grooves on the Teflon sheets, homeotropic ordering of cylindrical polymers may be nucleated and gradually propagate upon hot-pressing towards the interior of the film (Fig. 1b). The systematic study presented herein may provide a promising molecular design strategy for polymers that align homeotropically in a 2D plane.

## Results

**Molecular structures and thermal properties of polymers.** A bottlebrush polymer is a general term to describe a densely grafted polymer with long side chains, which can primarily be prepared by the polymerization of monomers with long side chains called macromonomers[36]. Because of a steric hindrance between its densely grafted side chains, the bottlebrush polymer backbone is forced to adopt a fully extended conformation and, as a result, the polymer adopts a cylindrical structure[37,38]. In the present study, we first engineered 13 bottlebrush polymers by combining three different backbones such as polymethacrylate (**PMA**), polyacrylate (**PA**) and polyphenylacetylene (**PPA**) with

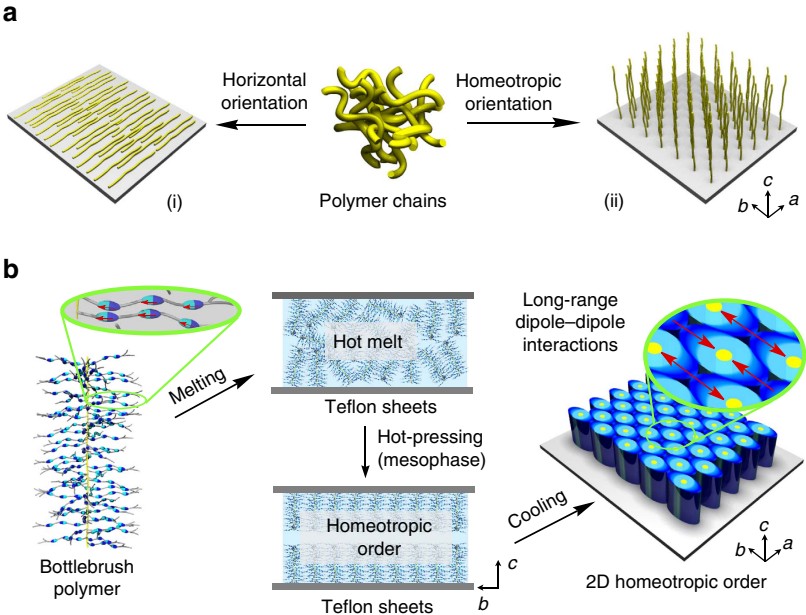

**Figure 1 | Design principles for polymers processable into a 2D homeotropic order.** (**a**) Schematic representations of the horizontal (i) and homeotropic (ii) orientations of polymer chains on a substrate. (**b**) Schematic representations of the self-assembly of a cylindrical bottlebrush polymer into a 2D homeotropic order. The polymer carries three polarized mesogenic units in its individual side chains and self-assembles into a 2D rectangular geometry, where constituent cylinders are deformed to have an ellipsoidal cross-section featuring oppositely oriented local dipoles. The interaction between these dipoles forces the cylinders to tightly connect bilaterally. With a physical assistance of the surface grooves on the Teflon sheets, nucleation for homeotropic ordering can be induced and propagate efficiently upon hot-pressing towards the interior of the film, wherein consistent polymer molecules align homeotropically.

three different side-chain mesogens such as biphenyl (**B**), tolan (**T**) and azobenzene (**A**; Fig. 2a). Bottlebrush polymers with **PMA** and **PA** as the backbones were obtained by free-radical polymerization of the corresponding macromonomers, whereas those with a **PPA** backbone were obtained by olefin metathesis polymerization using Rh(nbd)BPh$_4$ as a catalyst[39,40]. For the purpose of investigating possible effects of dipoles in the side chains, we prepared **PMA$^{BBB'}$** and **PMA$^{TTT'}$** as references where the outermost mesogenic unit (M3 in Fig. 2a) is connected at its both sides by ether oxygen atoms unlike other cases such as **PMA$^{BBB}$** and **PMA$^{TTT}$**, where an electronically push–pull structure is utilized for the incorporation of all mesogenic units (Fig. 2b). Detailed synthetic procedures and yields, degrees of polymerization and polydispersity indexes of all bottlebrush polymers synthesized are summarized in Supplementary Methods and Supplementary Table 1.

Phase diagrams of the individual bottlebrush polymers, as determined using differential scanning calorimetry (Supplementary Fig. 1 and Supplementary Table 2) and X-ray diffraction (Supplementary Figs 2–15) analysis, are summarized in Fig. 2c. As in the case of **PMA$^{AAA}$**, which was extensively studied in our previous work[35], all of the newly prepared bottlebrush polymers except **PA$^{BBB}$** and **PMA$^{BBB'}$** displayed a mesophase (Supplementary Fig. 1a–m). It is worthy of noting that the phase transition behaviours of the bottlebrush polymers are mainly determined by the types of mesogens rather than the type of the polymer backbone. For example, **PMA$^{BBB}$**, **PA$^{BBB}$** and **PPA$^{BBB}$** upon cooling from their isotropic melt underwent a phase transition at nearly the same temperature (Supplementary Fig. 1a–c), whereas the mesophase temperature range of **PMA$^{AAA}$** carrying the azobenzene side chains exclusively ($120 - 103\,°C$) was higher than those of **PMA$^{TTT}$** ($105 - 92\,°C$, see Supplementary Fig. 1d) and **PMA$^{BBB}$** ($104 - 99\,°C$, see Supplementary Fig. 1a). Accordingly, bottlebrush polymers such as **PMA$^{TTA}$**, **PMA$^{BBA}$**, **PMA$^{TAA}$** and **PMA$^{BAA}$**, which contain at least one azobenzene unit, displayed a higher mesophase temperature range than those having no azobenzene such as **PMA$^{TTT}$** and **PMA$^{BBB}$** (Supplementary Fig. 1e–h).

**2D homeotropic ordering of polymers upon hot-pressing.** By means of small-angle X-ray scattering (SAXS), we first investigated whether the bottlebrush polymers intrinsically adopt any ordered structure in bulk. As in the case of **PMA$^{AAA}$**, all of the bottlebrush polymers listed in Fig. 2b adopt, both in the mesophase and solid state, an ordered structure with a 2D rectangular lattice when they carry three ester-linked mesogenic units in their individual side chains. Of further note on the **PMA** family, the 2D rectangular lattices of **PMA$^{AAA}$** and **PMA$^{BBB}$** (Fig. 3a, Supplementary Fig. 2 and Supplementary Table 3) and those of their hybrids such as **PMA$^{BAA}$** (Supplementary Fig. 3 and Supplementary Table 4) and **PMA$^{BBA}$** (Supplementary Fig. 4 and Supplementary Table 5) commonly belong to the symmetry group of $P2_1/a$ irrespective of the ratio of mesogens **A** to **B**. Meanwhile, the 2D rectangular lattice of **PMA$^{TTT}$** belongs to the symmetry group of $C2/m$ (Fig. 3b, Supplementary Fig. 5 and Supplementary Table 6) and is different from the above series having mesogens **A** and/or **B** exclusively. However, when mesogen **A** or **B** is combined with mesogen **T** for the side-chain motif, the preferred symmetry group of the resulting 2D rectangular lattice is determined by the major mesogen. Namely, **PMA$^{TAA}$** (Supplementary Fig. 6 and Supplementary Table 7) and **PMA$^{TTA}$** (Supplementary Fig. 7 and Supplementary Table 8) prefer $P2_1/a$ and $C2/m$, respectively. Accordingly, irrespective of the sequence of three mesogenic units in the side chains,

**PMA$^{TTB}$**, **PMA$^{TBT}$** and **PMA$^{BTT}$**, the same as **PMA$^{TTA}$**, all prefer $C2/m$ because **T** is the major mesogen in these three bottlebrush polymers (Supplementary Figs $8 - 10$ and Supplementary Tables $9 - 11$). Just for curiosity, we replaced the ester linkages for connecting the outermost mesogenic units in **PMA$^{BBB}$** and **PMA$^{TTT}$** with an ether unit. Of interest, resulting **PMA$^{BBB'}$** (Fig. 3c, Supplementary Fig. 11 and Supplementary Table 12) and **PMA$^{TTT'}$** (Fig. 3d, Supplementary Fig. 12 and Supplementary Table 13) no longer adopts a 2D rectangular lattice but a 2D hexagonal lattice ($P6mm$). In sharp contrast, polymers with less than three mesogenic units in their side chains such as **PMA$^{BB}$**, **PMA$^{TT}$**, **PMA$^{B}$** and **PMA$^{T}$** gave ill-defined structures (Supplementary Figs 1n–q and 13).

Next, we hot-pressed these bottlebrush polymers between two Teflon sheets. Each Teflon sheet on its one side possesses unidirectionally oriented surface grooves. As in our previous study[35], the Teflon sheets were arranged such that their surface grooves were parallel to one another for sandwiching the bottlebrush polymers. Individual samples were allowed to cool from their isotropic melts to a temperature lower by $5\,°C$ than the phase transition temperature to the ordered phase and pressed at $8.0\,MPa$, and then slowly cooled to room temperature. Consequently, 6–10 μm-thick self-standing films were obtained (see Methods). In a through-view 2D SAXS image of the hot-pressed **PMA$^{BBB}$** ($P2_1/a$) film, arc-shaped diffractions assignable to the (110), (210) and (020) planes were clearly observed (Fig. 4a), suggesting that the (001) plane of the 2D lattice is oriented parallel to the surface of the Teflon sheets. Furthermore, the diffraction spots indexed to the (020) plane appeared only in a parallel direction to the surface grooves on the Teflon sheets. Consistent with the results of through-view 2D SAXS imaging, only (020) spots were observed in the equatorial direction of the edge-view 2D SAXS image (Supplementary Fig. 17a). These results clearly indicate that the $b$ axis of the 2D lattice, as illustrated in Fig. 4a (lower), is oriented parallel to the surface grooves on the Teflon sheets. In other words, **PMA$^{BBB}$** adopts a homeotropic order in the hot-pressed film. Consistently, in polarizing optical microscopy (POM), hot-pressed **PMA$^{BBB}$** under crossed polarizers exhibited a contrast at every $45°$ on rotation, giving bright and dark views when the azimuthal angles between the polarizing direction of the incident light and the $b$ axis of the 2D lattice were $45°$ and $0°$ ($90°$), respectively (Fig. 5a). Likewise, polarized infrared spectroscopy of the film gave polar plots with a clear dichroic feature (Fig. 5b) in which the stretching vibration bands attributable to the aromatic ether ($C_{Ar}$–O) and ester (C–O) groups, as well as those of the aromatic rings ($C_{Ar}$–$C_{Ar}$), of the mesogens displayed a maximum absorbance in the direction parallel ($0°$) to the surface grooves on the Teflon sheets. The same held true for the hot-pressed **PMA$^{TTT}$** in 2D X-ray diffraction (Fig. 4b), although its preferred symmetry group of the 2D lattice was $C2/m$ and different. Again, POM (Fig. 5c) and polarized infrared (Fig. 5d) unambiguously supported the homeotropic order of cylindrical **PMA$^{TTT}$** in the hot-pressed film. As shown by a time-dependent 2D SAXS imaging in Supplementary Fig. 21, this homeotropic ordering developed rather sluggishly and required **PMA$^{TTT}$** to be kept hot-pressed for several hours to accomplish. Meanwhile, shortly after the sample was hot-pressed, almost no structuring resulted.

Most importantly, not only **PMA$^{BBB}$** and **PMA$^{TTT}$** but also all the other **PMA**-based bottlebrush polymers, adopting a rectangular 2D lattice, displayed the same orientation behaviour upon being hot-pressed in the Teflon sheets. Namely, such cylindrical polymers align homeotropically with respect to the Teflon sheets when their surface grooves are arranged parallel to one another (Supplementary Figs 16, 17b–i and $23 - 27$ and Supplementary Table 16). In contrast, when bottlebrush polymers such as

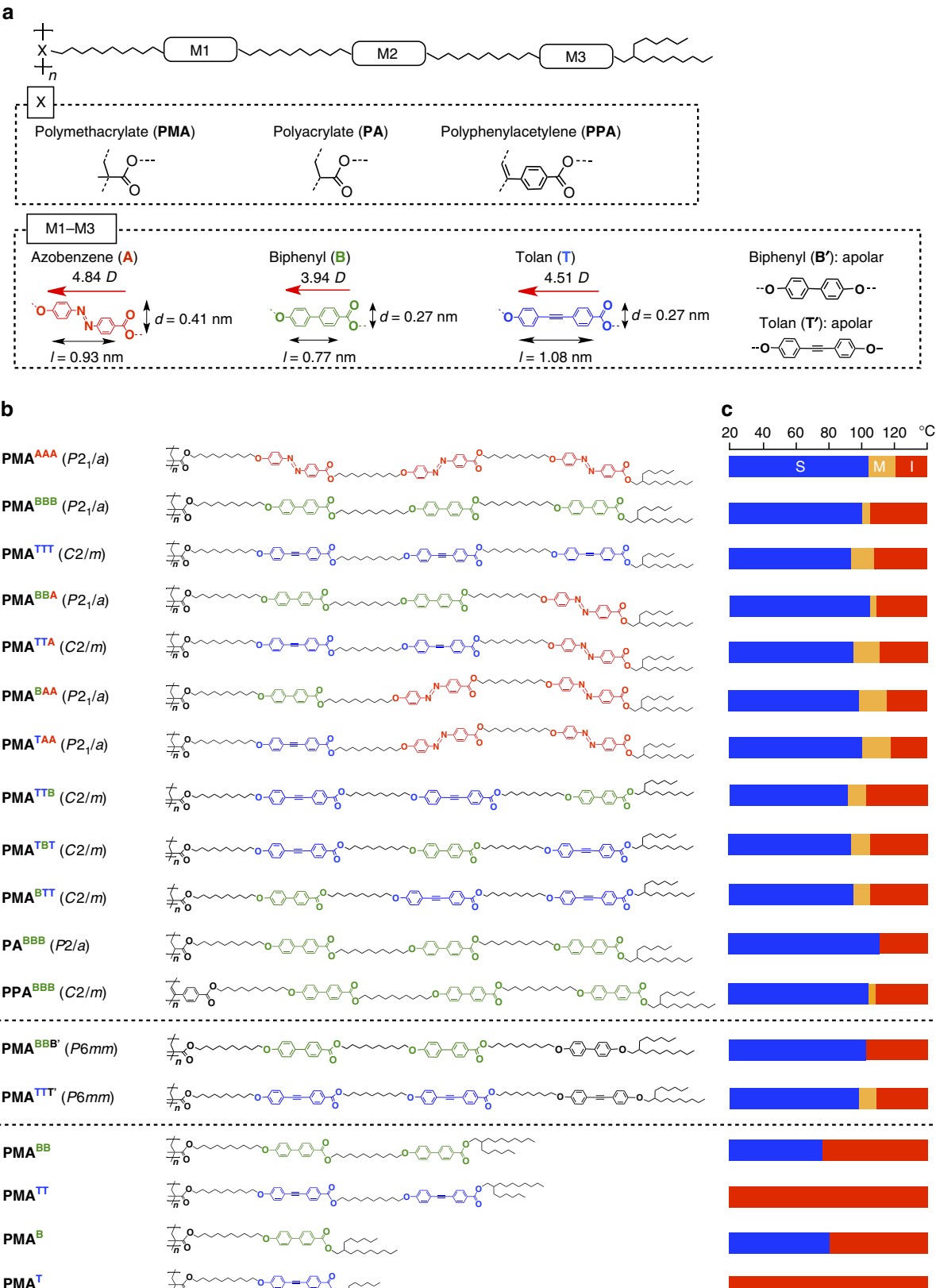

**Figure 2 | Molecular structures of bottlebrush polymers and their phase transition behaviours.** (**a**) Schematic representations of the molecular structures of newly developed bottlebrush polymers. All of the polymers contain mesogenic units M1–M3 in their individual side chains. Mesogenic units are shown in the inset, where *l* and *d* refer to the molecular lengths of a mesogen along its long and short axes, respectively. Red arrows denote the directions of the dipole moments of the mesogens. (**b**) Schematic molecular structures of bottlebrush polymers. (**c**) Differential scanning calorimetry traces for the phase transition behaviours of polymers in **b** upon cooling (scan rate; 5 °C min$^{-1}$). Red, orange and blue blocks denote the isotropic state (I), mesophase (M) and solid state (S) of individual polymers, respectively.

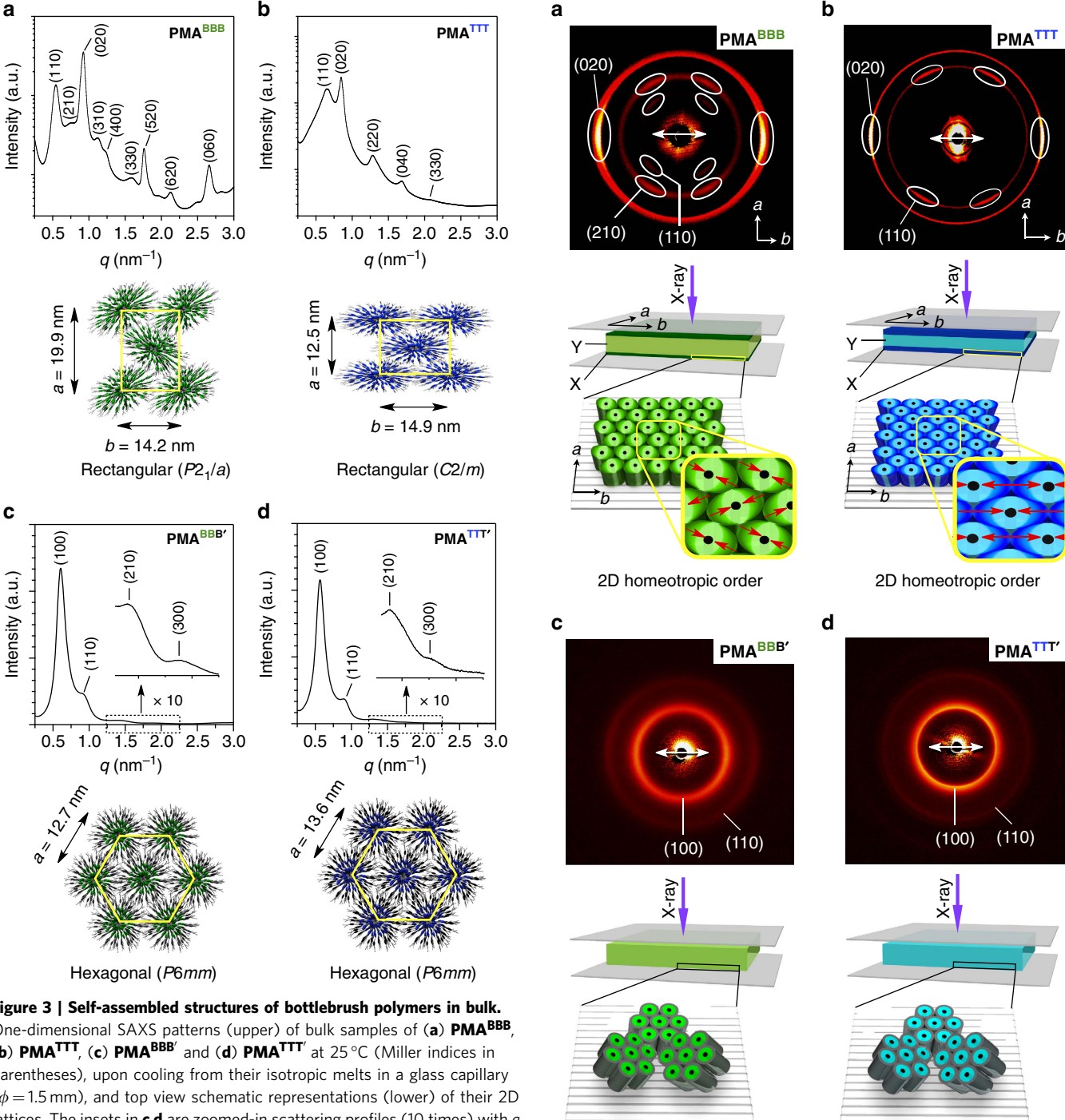

**Figure 3 | Self-assembled structures of bottlebrush polymers in bulk.** One-dimensional SAXS patterns (upper) of bulk samples of (**a**) **PMA**$^{BBB}$, (**b**) **PMA**$^{TTT}$, (**c**) **PMA**$^{BBB'}$ and (**d**) **PMA**$^{TTT'}$ at 25 °C (Miller indices in parentheses), upon cooling from their isotropic melts in a glass capillary ($\phi = 1.5$ mm), and top view schematic representations (lower) of their 2D lattices. The insets in **c,d** are zoomed-in scattering profiles (10 times) with $q$ values from 1.25 to 2.25 nm$^{-1}$.

**PMA**$^{BBB'}$ and **PMA**$^{TTT'}$, which adopt a hexagonal columnar lattice, were likewise hot-pressed in the Teflon sheets, only isotropic circles emerged in their through-view 2D X-ray diffraction profiles (Fig. 4c,d, upper). Together with the results of POM (Supplementary Fig. 22) and polarized infrared (Supplementary Fig. 28), these 2D X-ray diffraction profiles clearly indicate the absence of any macroscopic structural order in the hot-pressed **PMA**$^{BBB'}$ and **PMA**$^{TTT'}$ films (Fig. 4c,d, lower). Despite such a high sensitivity to the side-chain structure, the polymer backbone does not play an essential role. In fact, analogous to the **PMA** series, **PA**$^{BBB}$ and **PPA**$^{BBB}$ having **PA** and **PPA** backbones, respectively, adopt a 2D

**Figure 4 | Orientation of bottlebrush polymers in their hot-pressed films.** Through-view 2D SAXS images (upper) at 25 °C of hot-pressed films of (**a**) **PMA**$^{BBB}$, (**b**) **PMA**$^{TTT}$, (**c**) **PMA**$^{BBB'}$ and (**d**) **PMA**$^{TTT'}$, and schematic representations (lower) of their molecular arrangements. The surface grooves on the Teflon sheets are depicted by black lines, whose directions are highlighted by white arrows. Red arrows denote the directions of the oppositely oriented local dipoles in the side chains. X and Y indicate the homeotropic ordered and disordered domains in hot-pressed films, respectively.

rectangular lattice in bulk (Supplementary Figs 14 and 15 and Supplementary Tables 14 and 15) and align homeotropically in their hot-pressed films (Supplementary Fig. 18).

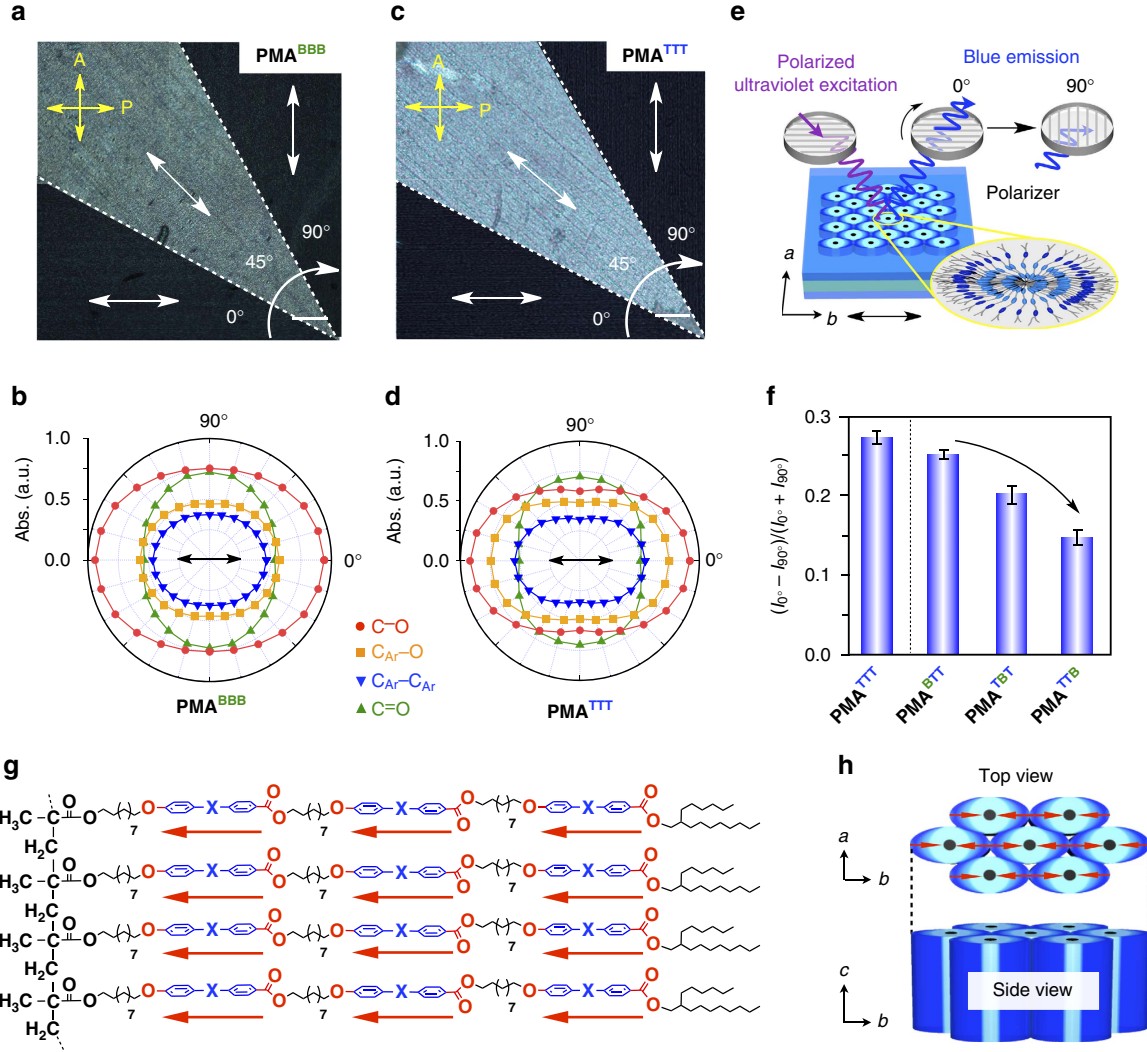

**Figure 5 | Anisotropic orientation of side-chain mesogenic units of bottlebrush polymers in their hot-pressed films.** (**a,c**) POM micrographs of hot-pressed films of (**a**) **PMA**[BBB] and (**c**) **PMA**[TTT] under crossed polarizers, recorded at 0° (lower regions), 45° (diagonal regions) and 90° (upper regions) relative to the transmission axis of the polarizer (P, yellow arrow) upon clockwise rotation of the film (white circular arrow). Scale bars, 50 µm. White arrows in all of the images denote directions of the surface grooves on the Teflon sheets. (**b,d**) Polar plots of the infrared absorption intensities, recorded upon rotation of a polarizer at every 15°, of hot-pressed films of (**b**) **PMA**[BBB] and (**d**) **PMA**[TTT]. The azimuthal angle is defined as 0° when the polarizing direction of the incident light is parallel to the surface grooves on the Teflon sheets (black arrows). (**e**) Schematic representation of anisotropic fluorescence experiments and orientation of mesogens in the side chains (top view). When a film sample is excited by polarized ultraviolet light (310 nm, purple line) parallel to the surface grooves on the Teflon sheets (black arrows), a polarized blue emission (420 nm, blue line) appears. The outermost mesogens are oriented more parallel to the surface grooves on the Teflon sheets than those of inner mesogens. (**f**) Fluorescence anisotropy $(I_{0°} - I_{90°})/(I_{0°} + I_{90°})$ of hot-pressed films of **PMA**[TTT], **PMA**[BTT], **PMA**[TBT] and **PMA**[TTB], where $I_{0°}$ and $I_{90°}$ are the fluorescence intensities in the polarized emission spectra when the excitation and emission polarizers form angles of 0 and 90°, respectively. Error bars represent s.d. (**g**) Schematic representation of local dipoles formed by the ester and ether groups in the individual side chains of a bottlebrush polymer examined in the present study. (**h**) Schematic representations of a 2D assembly of bottlebrush polymer molecules with an ellipsoidally deformed cross-section into a rectangular lattice via a dipole–dipole interaction. Red arrows denote the directions of the oppositely oriented local dipoles in the side chains.

**Orientation of mesogens in polymers at their cross-section.** Polarized fluorescence spectroscopy is a useful tool for investigating the orientation of fluorophores[41,42]. The fluorescent intensity of **PMA**[TTT] under ultraviolet irradiation was sufficiently strong to be detected by the naked eye, even in its film state, whereas **PMA**[BBB] was barely emissive (Supplementary Fig. 29). In fact, the fluorescence spectral patterns of **PMA**[BTT], **PMA**[TBT] and **PMA**[TTB] are almost identical to that of **PMA**[TTT] either in solution or in the film state, indicating that the contribution of mesogen **B** is negligibly small (Supplementary Figs 30 and 31). Therefore, comparison of the fluorescence anisotropies of the hot-pressed films of **PMA**[BTT], **PMA**[TBT] and **PMA**[TTB] possibly allows for evaluating the degrees of orientation of mesogen **T** at different positions, that is, M1, M2 and M3 (Fig. 2a) in the side chains. As explained schematically in Fig. 5e, we investigated the degrees of fluorescence anisotropy of three hot-pressed polymer films **PMA**[BTT], **PMA**[TBT] and **PMA**[TTB] using a polarized light parallel to the surface grooves on the Teflon sheets. As a result, when mesogen **B** is further displaced from the backbone (**PMA**[BTT]→**PMA**[TBT]→**PMA**[TTB]), the fluorescence anisotropy decreases from 0.25 (**PMA**[BTT]) to 0.20 (**PMA**[TBT]) and then to 0.15 (**PMA**[TTB]; Fig. 5f). These results

suggest that the outermost mesogen (M3) units, which are likely located in the least congested environment, align more parallel to the surface grooves on the Teflon sheets, whereas the innermost mesogen (M1) units are oriented more radially and therefore more isotropically (Fig. 5e).

## Discussion

Through the aforementioned systematic study using 14 bottle-brush polymers with triple-mesogenic long side chains (Fig. 2), we found that whether the polymer self-assembles into a 2D rectangular lattice ($PMA^{AAA}$–$PPA^{BBB}$ in Table 1) rather than a 2D hexagonal lattice ($PMA^{BBB'}$ and $PMA^{TTT'}$ in Table 1) is crucial for the homeotropic orientation of the polymer backbone upon hot-pressing with Teflon sheets. In the hot-pressed films of $PMA^{AAA}$–$PPA^{BBB}$, the *ab* plane of the 2D rectangular lattice is oriented parallel to the Teflon sheet surface with its *b* axis directed along the surface grooves. Furthermore, it is clear that the mesogenic units align along the surface grooves on the Teflon sheets, that is, the *b* axis of the 2D rectangular lattice, considering the anisotropic POM patterns of the hot-pressed films (Fig. 5a,c) and highly dichroic optical features (Fig. 5b,d) of their mesogenic groups. Consequently, just like the case of $PMA^{AAA}$ discussed before[35], the polymer backbones, which align along the *c* axis of the 2D rectangular lattice, are homeotropic with respect to the film plane. Some ordinary rod-like liquid crystalline molecules were reported to orient their mesogenic units along the surface grooves on the Teflon sheet[43,44]. Furthermore, the mesogenic units of the monomers of $PMA^{BBB}$ and $PMA^{TTT}$, as well as that of $PMA^{AAA}$ reported previously[35], were observed to align parallel to the surface grooves on the Teflon sheet (Supplementary Fig. 19). Hence, we consider that the surface grooves on the Teflon sheets nucleate the homeotropic orientation of bottlebrush polymers (Fig. 1b). A possible advantage of the 2D rectangular lattice would be its deformability to an ellipsoidal shape, which allows the constituent mesogenic units to align along the surface grooves as much as possible. Meanwhile, in a hexagonal columnar lattice, the constituent cylindrical polymer ($PMA^{BBB'}$ or $PMA^{TTT'}$) would possibly adopt an entropically favoured round shape with isotropically extended mesogenic units (Fig. 3c,d, lower). Hence, groove-directed unidirectional orientation of the mesogenic units, leading to the homeotropic

order of the bottlebrush polymers (Fig. 4c,d, lower), would hardly be realized here.

Then, what are the requisites for polymers to assemble into a 2D rectangular lattice? Considering the fact that $PMA^{BBB'}$ and $PMA^{TTT'}$ do not assemble into a 2D rectangular lattice but a 2D hexagonal lattice, we consider that the presence of three ester groups in each side chain plays an essential role in the formation of a 2D rectangular lattice. Taking $PMA^{TTT}$ as an example, all three mesogenic units in its individual side chains are polarized along the same direction by the attachment with an electron-withdrawing ester carbonyl group and an electron-donating ether oxygen atom (Fig. 5g). Consequently, its individual side chains bear a large dipole from the terminus to the backbone core. Further to note, $PMA^{TTT}$, which assembles into a 2D rectangular lattice, is deformed ellipsoidally in its cross-section, where oppositely oriented local dipoles are supposed to emanate from the terminus (Fig. 5e). Although these local dipoles are cancelled with one another within each cylinder, they can be locally interactive with those of neighbouring columns[45]. Namely, such ellipsoidally deformed cylinders are tightly connected bilaterally via a dipole–dipole interaction and form a 2D rectangular lattice (Fig. 5h). As described for the time-dependent 2D SAXS imaging in Supplementary Fig. 21, the homeotropic order of bottlebrush polymers such as $PMA^{TTT}$ develops rather gradually upon hot-pressing. Shortly, hot-pressed $PMA^{TTT}$ films adopt almost no particular orientation, suggesting that the homeotropic order develops with a thermodynamically controlled flavour. Note that the 2D rectangular lattice in such a hot-pressed film is constructed in such a way that its *b*- and *c* axes uniformly align parallel and perpendicular to the surface grooves on the Teflon sheets, respectively[35] (Fig. 1b).

Finally, we investigated order parameters of the hot-pressed films. Because the 2D assembly with a rectangular geometry is nucleated at the polymer/Teflon sheet interface, the order parameter of a bottlebrush polymer generally decreases as the film thickness increases. We noticed an interesting relationship between the thickness of the homeotropically ordered domain (X in Fig. 4a,b, lower) and total $\pi$-plane surface area of all the mesogens ($^{all}S_\pi$) in its individual side chains. Value X was estimated by integrating the scattering peak of the (020) plane in its 2D SAXS data (Supplementary Figs 32 and 33), while the total $\pi$-plane surface area $^{all}S_\pi$ was obtained in Table 1 by sum of the

## Table 1 | Structural parameters of polymers and their assemblies.

| Symbol | Polymers | Lattice parameters at 25 °C | | | Hot-pressed films | |
|---|---|---|---|---|---|---|
| | Total surface area $^{all}S_\pi$ (nm²)* | Space group[†] | *a* (nm) | *b* (nm) | X (μm)[‡] | Y (μm)[§] |
| $PMA^{AAA}$ | 1.2 | P2₁/a | 21.8 | 14.7 | 2.6 | 4.9 |
| $PMA^{BBB}$ | 0.6 | P2₁/a | 19.9 | 14.2 | 0.5 | 6.0 |
| $PMA^{TTT}$ | 0.9 | C2/m | 12.5 | 14.9 | 1.0 | 5.0 |
| $PMA^{BBA}$ | 0.8 | P2₁/a | 22.4 | 14.1 | 0.6 | 4.8 |
| $PMA^{TTA}$ | 1.0 | C2/m | 11.8 | 14.9 | 1.1 | 4.8 |
| $PMA^{BAA}$ | 1.0 | P2₁/a | 21.2 | 14.3 | 1.3 | 5.4 |
| $PMA^{TAA}$ | 1.1 | P2₁/a | 21.7 | 15.2 | 1.6 | 4.8 |
| $PMA^{TTB}$ | 0.8 | C2/m | 12.8 | 14.6 | 0.9 | 5.2 |
| $PMA^{TBT}$ | 0.8 | C2/m | 12.7 | 14.8 | 0.6 | 5.7 |
| $PMA^{BTT}$ | 0.8 | C2/m | 12.9 | 14.4 | 0.5 | 6.0 |
| $PA^{BBB}$ | 0.6 | P2/a | 13.9 | 12.4 | 0.8 | 4.4 |
| $PPA^{BBB}$ | 0.6 | C2/m | 12.6 | 14.2 | 0.6 | 4.8 |
| $PMA^{BBB'}$ | 0.6 | P6mm | 12.7 | — | Random orientation | |
| $PMA^{TTT'}$ | 0.9 | P6mm | 13.6 | — | Random orientation | |

2D, two-dimensional; PA, polyacrylate; PMA, polymethacrylate; PPA, polyphenylacetylene.
*$^{all}S_\pi$: total $\pi$-plane surface area of all mesogens involved in the individual side chains.
†P2₁/a, P2/a and C2/m: space groups of the 2D rectangular lattices; P6mm: space group of the 2D hexagonal lattice.
‡X: thickness of a homeotropic ordered domain in a hot-pressed film from each side.
§Y: thickness of a disordered domain in a hot-pressed film.

$\pi$-plane surface areas of all mesogens $^BS_\pi$ (0.21 nm$^2$), $^TS_\pi$ (0.29 nm$^2$) and/or $^AS_\pi$ (0.38 nm$^2$) involved in the side chains ($S_\pi = l \times d$, Fig. 2a). As shown in Supplementary Fig. 34, the plots of $X$ against $^{all}S_\pi$ clearly showed that the order parameter of the film increases with increasing $^{all}S_\pi$. In all of the bottlebrush polymers examined, **PMA$^{AAA}$** with the largest $^{all}S_\pi$ gave the highest-order parameter ($X = 2.5\,\mu$m), where the cylindrical polymer objects located in a 2.5-$\mu$m-thick area from the polymer/Teflon sheet interface were homeotropically oriented[35]. This trend indicates a large contribution of the $\pi$-stacking interaction along the cylindrical polymer axis to the orientational integrity of the hot-pressed film.

Through the present systematic study with 13 newly designed bottlebrush polymers (Fig. 2) along with our previous work[35] featuring the anomalous assembling behaviours of **PMA$^{AAA}$**, we extracted necessary structural and conditional elements for cylindrical bottlebrush polymers to develop a 2D homeotropic order upon hot-pressing in Teflon sheets: (1) three polarized mesogenic units in individual side chains, (2) intrinsically strong preference for the self-assembly into a 2D rectangular lattice, (3) less important polymer backbone structure, (4) a large total $\pi$-plane surface area of the mesogenic units and (5) a sufficiently long hot-pressing time (several hours). Individual cylindrical polymers have an ellipsoidally deformed cross-section with oppositely directed local dipoles that emanate from the terminus. The interaction of these local dipoles possibly forces neighbouring cylinders to tightly connect bilaterally. With a physical assistance of the surface grooves on the Teflon sheets, homeotropic structural ordering is likely nucleated and gradually propagates upon hot-pressing towards the interior of the film (Fig. 1b). Consequently, a 2D rectangular lattice is constructed in the resulting hot-pressed film in such a way that its $b$- (side chains) and $c$ axes (polymer backbone) uniformly align parallel and orthogonal to the surface grooves on the Teflon sheets.

Before our previous work on **PMA$^{AAA}$**, freestanding films composed of homeotropically oriented polymers were unprecedented[35]. Its underlying strategy and principle disclosed in the present systematic study may address an essential issue of how one can orient polymers homeotropically rather than horizontally. The importance of dipole–dipole interactions has been well understood for orienting small molecules but not for polymers because of a notion that local dipoles in polymers must be randomized[45–49]. However, this does not hold true for bottlebrush polymers with an extended backbone conformation. Since bottlebrush polymers have the ability to accommodate a variety of functional groups in a site-specific manner, the present work possibly contribute to the development of polymer-based 2D materials of anomalous functions.

## Methods

**Processing by hot-pressing.** A cast polymer film prepared from its CHCl$_3$ solution (1 mg ml$^{-1}$) was sandwiched by two anisotropic Teflon sheets (10 cm × 10 cm) such that their surface grooves were parallel to one another. The hot-pressing was conducted under 8.0 MPa in a mesophase temperature range for 1 h (5 °C lower than the phase transition temperature from the isotropic melt to an ordered phase) after heating to an isotropic state shortly (10 °C higher than the melting temperature) for complete melting. Then, the sample was allowed to cool to 25 °C. A self-standing polymer film was peeled from the Teflon sheet using a blade. Note that thermally annealed films of **PMA$^{BBB}$** and **PMA$^{TTT}$** on a Teflon sheet after drop-casting without pressing did not exhibit any 2D SAXS feature representing their homeotropic orientation (Supplementary Fig. 20), although the films were slightly birefringent along the surface grooves on the Teflon sheet. Therefore, it is likely that the surface nucleation for the homeotropic polymer orientation may not occur without pressing, but the pressing would assist the propagation of the surface nucleation efficiently towards the interior of the film sample.

**Synchrotron X-ray diffraction analysis.** Powdered samples were placed into a 1.5 mm-$\phi$ glass capillary in a temperature-controlled heating block and heated to form an isotropic melt. Then, the resulting samples were allowed to cool to 25 °C and were exposed to an X-ray beam for 10 (WAXD) or 100 (SAXS) seconds at given temperatures. Film samples were clipped with tweezers and exposed to an X-ray beam for 300 s at 25 °C. For experimental details for these analyses and synthesis of compounds, see the Supplementary Methods and Discussions.

**Data availability.** All relevant data are included in full within this paper and in the Supplementary Information.

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

## Acknowledgements

We acknowledge a JSPS Grant-in-Aid for Specially Promoted Research (25000005) on 'Physically Perturbed Assembly for Tailoring High-Performance Soft Materials with Controlled Macroscopic Structural Anisotropy'. We also acknowledge the ImPACT Program of the Council for Science, Technology and Innovation (Cabinet Office, Government of Japan). The synchrotron X-ray diffraction experiments were performed at BL45XU in SPring-8 with the approval of the RIKEN SPring-8 Center (proposal 20110065, 20120045 and 20130025).

## Author contributions

Z.C. designed and performed all experiments. Y.-T.C., T.K. and D.M. co-designed the experiments. J.M.L. and A.K. performed partial synthetic experiments. Z.C., T.K., D.M., T.F. and T.A. analysed the data and wrote the manuscript.

## Additional information

**Competing financial interests**: The authors declare no competing financial interests.

**How to cite this article**: Chen, Z. *et al.* A design principle of polymers processable into 2D homeotropic order. *Nat. Commun.* **7**, 13640 doi: 10.1038/ncomms13640 (2016).

**Publisher's note**: 

