## [Peer Review File · Nature Communications]

Reviewers' Comments:

Reviewer #1 (Remarks to the Author)

The manuscript describes an innovative new strategy for manipulating molecular structure in such a way that cylindrical (or oblate) polymers spontaneously organize with the long axis of the cylinder (or oblate cylinder) perpendicular to a surface. The resulting polymeric films have potential to significantly improve morphological control for polymeric materials used as membranes that support transport of ions, energy, or small molecules. I expect that these results will be very influential.

The experimental work is extremely well done. The authors have assembled a large library of structurally related polymers. All of the compounds (including intermediates) are thoroughly characterized as detailed in the extensive supporting information file. Characterization by differential scanning calorimetry, small- and wide-angle x-ray scattering, and polarized optical microscopy offer convincing evidence that the polymers are indeed homeotropically aligned. By virtue of the size of the compound library studied, the authors are able to show the generality of the molecular design principle. Most remarkable, though, is that insights about the underlying mechanism that causes homeotropic alignment implied by variations in the molecular structures are substantiated through microscopic, scattering, and polarized IR and fluorescence spectroscopy. From this exhaustive study the authors are well positioned to advance a model to rationalize the observation of robust homeotropic alignment of their polymers.

Briefly, the polymers investigated in this work are composed of a linear polymer backbone and macromolecular side chains. The large size of the side chains gives the individual polymer chains cylindrical (or oblate) shape with a nanometer-scale diameter. The authors show that the cylindrical polymers stand vertically from the grooved Teflon surface after hot press processing. Furthermore, polarizable groups in the side chains are shown to have anisotropic distribution about the cylinder axis. Based on structure-property relationships within the compound library, the authors note that polarizable groups at the cylinder surface are needed to achieve homeotropic alignment. This leads the authors to propose that the dipoles of these surface groups are interacting to promote coaxial alignment of the oblate cylinders. What is not clear is why homeotropic, rather than planar, orientation is nucleated when these terminal polar groups are present.

The mechanism described in this manuscript is unique from prior work on homeotropic alignment of cylindrical polymers. In addition to the work on conjugated and graft polymers referred to in the manuscript, previous work on dendronized polymers, which have a similar bottlebrush architecture, focused on surface energy mismatch to promote homeotropic alignment (uncited references: *Nature* 2002, 419, 384 & *Macromolecules* 2015, 48, 2849). These previous molecular design strategies have not proven to be as robust as the one described in the present work.

A few issues caught my attention while reading the manuscript.

1. On pages 3 and 9 the authors use the phrases "structurally elaborated polymers" and "structurally elaborate bottlebrush polymers." I'm not sure what "elaborated" or "elaborate" mean in this context. Maybe there's a better expression to use.
2. The description of bottlebrush polymers on page 4 is not very helpful to a broad audience because you use the term "bottlebrush" to describe a bottlebrush architecture.
3. In the design requirements described on page 11 (lines 279-281), the authors state that the macromolecular side chains must contain three polarizable mesogens. This is based on the absence of a mesophase in polymers with shorter side chains (e.g., PMA(BB), PMA(TT), PMA(B)

and PMA(T)), and the absence of homeotropic alignment in samples where the terminal group is a non-polarizable mesogen (e.g., PMA(BBB') and PMA(TTT')). The generalization on page ten suggests that the hypothetical bottlebrush polymers PMA(B'BB) and PMA(B'B'B) would not homeotropically align. However, the model advanced in the manuscript does not justify such an expectation. I look forward to seeing whether polymers such as PMA(B'BB) and PMA(B'B'B) homeotropically align.

4. Figure 1 caption (page 20, line 475) like has a misspelling "stricture" instead of "structure."

5. Page 11, line 288: "constituents polymer molecules" should be "constituent..."

6. Page 3, line 69: "ground challenge" should be "grand challenge."

7. In the supporting information file, the authors use an unusual notation in some of the NMR spectral data. Examples can be found on pages S7-S9. The notations in question are "d x 2", "d x 3", "t x 2", and "t x 3" for multiplicity of peaks. Maybe this can be explained with one sentence in the methods section.

8. On page S36 of the supporting information, there are two instances where "flack" was written instead of "flask."

Reviewer #2 (Remarks to the Author)

The manuscript by Chen et al. details the organization of a series of side-chain mesogenic polymers on Teflon sheets. The authors consider a broad combination of mesogen sequences as well as backbone chemistries, 14 samples in total, and assess the orientation of the columnar structures formed by the polymer chains (bottlebrushes) when hot pressed between Teflon sheets. The authors find that in all but two cases that homeotropic anchoring of the system is recovered. The two cases which do not show homeotropic anchoring form present P6mm symmetry, whereas the homeotropic cases are of different symmetries. These two exceptions differ principally by the presence of an ether bond, rather than an ester bond, connecting the alkyl tail to the aromatic core of the terminal mesogen in the side chain. On this basis the authors assert that homeotropic anchoring is reliant on the presence of sufficiently strong inter-columnar dipole interactions which lead to the formation of a 2D rectangular lattice/tight bilateral connections, resulting in large sheet like assemblies that prefer to lie flat on a substrate.

The data reported here represents something of a tour de force in terms of experimental effort. They represent the results of a systematic series of high quality experimental investigations. However the conclusions drawn by the authors seem rather premature - the connection between dipole interactions and surface anchoring is tenuous: while one may accept that strong dipole-dipole interactions lead to large correlation lengths (i.e. "...large sheet like assembly..."), this is a different matter than accounting for the orientation of such an assembly at a given surface. Further, there are gaps/omissions in the data that preclude arrival at a firm conclusion regarding the origin of the differences in the anchoring condition:

1. The role of the pressing temperature/rate and of the surface grooves of the Teflon substrate have not been considered. For the purposes of investigating what molecular aspects contribute to planar vs homeotropic anchoring, the presence of potentially strong shear flows as well as topographical variation of the substrate are non-trivial confounding factors. We have no way of knowing whether the observed alignment is in fact one driven by shear flow and/or topography, rather than by some other effects. As pointed out above, a priori one does not have a reason to expect that dipole interactions would affect anchoring conditions, moreso than say just correlation length, so one is left with more questions than answers in this respect.

2. There is no cross-sectional scattering or TEM provided. The 4-fold symmetry in the POM could easily result from near-surface effects that may or may not be uniformly distributed across the

systems.

3. The dipole moment is not quantified (eg. by AFM measurements, or molecular modeling). Beyond these issues, the paper as constructed does not do a good job of communicating the results presented. For example the discussion on p11 regarding the pi-plane area of the mesogens was very difficult to follow. No consideration is given to the issue of molecular weight, not of flow alignment/other effects, as mentioned above. The arguments in the introduction regarding the need for homeotropic anchoring of polymers at surfaces are not well formulated - in reality the desire for homeotropic anchoring is generally in self-assembled systems where one seeks vertical orientation of self-assembled structures, rather than individual polymer chains. Few readers will resonate with the idea that engineering homeotropic anchoring of polymer chains is a topic of intense concern. Further, while the introduction suggests that a general strategy has been formulated for controlling polymer chain organization, in reality the work here has uncovered differences in the organization of columnar superstructures formed by large mesogenic side chains in a liquid crystalline bottlebrush system, when hot-pressed between grooved Teflon sheets, i.e. not a general strategy/phenomenon, but a highly specific scenario, which as discussed above, is not well rationalized from a fundamental perspective.

For these reasons, publication in Nature Communication is not recommended.

Reviewer #3 (Remarks to the Author)

Summary of the key results

The authors present an efficient method to orient polymers on substrates homeotropically. In 2010, they found that a bottlebrush polymer bearing three azobenzene units in their side chains could 'stand' on the substrate (Science 330, 808). After a systematic study described in this manuscript, they reached a conclusion that the interaction between local dipoles in the side chains forces 'itches' from neighboring cylindrical polymers to form a large 2D sheet-like assembly, and then contributed to the homeotropical alignment of polymer backbones. All the experiments are well designed and performed and this paper deserves an urgent publication in Nature Communication.

Originality and interest: Novel

Data & methodology: Well-designed strategy and high quality of data presented in scientific way
quality of presentation

Appropriate use of statistics and treatment of uncertainties: Yes

Conclusions: Reliable

Suggested improvements: No revision

References: appropriate credit to previous work? yes

[3] Point-to-Point Answers to Reviewers' Comments

For Reviewer 1:

- (1) The manuscript describes an innovative new strategy for manipulating molecular structure in such a way that cylindrical (or oblate) polymers spontaneously organize with the long axis of the cylinder (or oblate cylinder) perpendicular to a surface. The resulting polymeric films have potential to significantly improve morphological control for polymeric materials used as membranes that support transport of ions, energy, or small molecules. I expect that these results will be very influential.

=> We highly appreciate these encouraging remarks.

- (2) The experimental work is extremely well done. The authors have assembled a large library of structurally related polymers. All of the compounds (including intermediates) are thoroughly characterized as detailed in the extensive supporting information file. Characterization by differential scanning calorimetry, small- and wide-angle x-ray scattering, and polarized optical microscopy offer convincing evidence that the polymers are indeed homeotropically aligned. By virtue of the size of the compound library studied, the authors are able to show the generality of the molecular design principle. Most remarkable, though, is that insights about the underlying mechanism that causes homeotropic alignment implied by variations in the molecular structures are substantiated through microscopic, scattering, and polarized IR and fluorescence spectroscopy. From this exhaustive study the authors are well positioned to advance a model to rationalize the observation of robust homeotropic alignment of their polymers.

=> We highly appreciate these encouraging remarks.

- (3) Briefly, the polymers investigated in this works are composed of a linear polymer backbone and macromolecular side chains. The large size of the side chains gives the individual polymer chains cylindrical (or oblate) shape with a nanometer-scale diameter. The authors show that the cylindrical polymers stand vertically from the grooved Teflon surface after hot press processing. Furthermore, polarizable groups in the side chains are shown to anisotropic distribution about the cylinder axis. Based on structure-property relationships within the compound library, the authors note that polarizable groups at the cylinder surface are needed to achieve homeotropic alignment. This leads the authors to propose that the dipoles of these surface groups are interacting to promote coaxial alignment of the oblate cylinders. What is not clear is why homeotropic, rather than planar, orientation is nucleated when these terminal polar groups are present.

- => In our previous paper (*Science*, **2010**, 330, 808), we reported that the monomer for **PMA^{AAA}**, when hot-pressed with Teflon sheets, aligns along the grooves of the Teflon sheets. We also reported that **PMA^{AAA}** in its 2D rectangular lattice intrinsically orients its mesogenic side chains mainly along the *b*-axis. Consequently, upon hot-pressing, polymer domains of **PMA^{AAA}** are unidirectionally oriented in such a way that the *ab* planes of their individual 2D rectangular lattices align parallel to the surface of the Teflon sheets. Hence, their *c*-axes align homeotropically with respect to the Teflon sheet surface. This represents how the homeotropic orientation of **PMA^{AAA}** is nucleated. Also for the purpose of addressing one of the comments raised by Reviewer 2, we added to the revised manuscript elaborate descriptions on this orientation mechanism (page 9, lines 10–19).
- => The important message of the present work is that the same orientation mechanism is applicable to other newly synthesized bottlebrush polymers (Fig. 4 and Supplementary Fig. 16) as well as their monomers (Supplementary Fig. 19).
- (4) The mechanism described in this manuscript is unique from prior work on homeotropic alignment of cylindrical polymers. In addition to the work on conjugated and graft polymers referred to in the manuscript, previous work on dendronized polymers, which have a similar bottlebrush architecture, focused on surface energy mismatch to promote homeotropic alignment (uncited references: *Nature* **2002**, 419, 384 & *Macromolecules* **2015**, 48, 2849). These previous molecular design strategies have not proven to be as robust as the one described in the present work.
- => The revised version now includes two suggested papers as references 33 and 34.
- (5) A few issues caught my attention while reading the manuscript. On pages 3 and 9 the authors use the phrases “structurally elaborated polymers” and “structurally elaborate bottlebrush polymers.” I’m not sure what “elaborated” or “elaborate” mean in this context. Maybe there's a better expression to use.
- => The phrases “structurally elaborated polymers” and “structurally elaborate bottlebrush polymers” were revised as “newly designed polymers” and “newly designed bottlebrush polymers”, respectively.
- (6) The description of bottlebrush polymers on page 4 is not very helpful to a broad audience because you use the term “bottlebrush” to describe a bottlebrush architecture.
- => The original sentence “A bottlebrush polymer is a general term to describe a graft polymer with a bottlebrush structure” was revised as “A bottlebrush polymer is a general term to describe a densely grafted polymer with long side chains”.

(7) In the design requirements described on page 11 (lines 279–281), the authors state that the macromolecular side chains must contain three polarizable mesogens. This is based on the absence of a mesophase in polymers with shorter side chains (e.g., PMA^{BB} , PMA^{TT} , PMA^{B} and PMA^{T}), and the absence of homeotropic alignment in samples where the terminal group is a non-polarizable mesogen (e.g., PMA^{BBB} and PMA^{TTT}). The generalization on page ten suggests that the hypothetical bottlebrush polymers $\text{PMA}^{\text{B'BB}}$ and $\text{PMA}^{\text{B'B'B}}$ would not homeotropically align. However, the model advanced in the manuscript does not justify such an expectation. I look forward to seeing whether polymers such as $\text{PMA}^{\text{B'BB}}$ and $\text{PMA}^{\text{B'B'B}}$ homeotropically align.

=> We actually spent a rather long time to synthesize reference polymers such as $\text{PMA}^{\text{B'BB}}$ and $\text{PMA}^{\text{B'B'B}}$ and/or PMA^{TTT} and $\text{PMA}^{\text{T'T'T}}$, since we really wanted to know their orientation behaviors upon hot-pressing. However, we could not establish appropriate synthetic methods for their monomers having inner ether units without unfavorable polymerization and finally gave up in our previous submission. Considering the comment of this reviewer, we decided to tone down the following rather strong message “Such polymers should adopt a cylindrical bottlebrush structure, whose individual side chains must include, at least, three polarized mesogenic units, each of which is attached to an electron-withdrawing ester carbonyl group and an electron-donating ether oxygen atom” from the manuscript. The revised sentence now is “Such polymers are considered to adopt a cylindrical bottlebrush structure, whose individual side chains should supposedly include three polarized mesogenic units just like the structure shown in Fig. 5g” (page 12, lines 3–5).

(8) Figure 1 caption (page 20, line 475) like has a misspelling “stricture” instead of “structure”.

(9) Page 11, line 288: “constituents polymer molecules” should be “constituent...”

(10) Page 3, line 69: “ground challenge” should be “grand challenge”.

=> We appreciate these suggestions. These typos were corrected in the revised version.

(11) In the supporting information file, the authors use an unusual notation in some of the NMR spectral data. Examples can be found on pages S7–S9. The notations in question are “d x 2”, “d x 3”, “t x 2”, and “t x 3” for multiplicity of peaks. Maybe this can be explained with one sentence in the methods section.

=> The revised “Supplementary Information” does not include the unusual notations suggested by the reviewer but utilizes a conventional method for assigning signals whenever possible.

(12) On page S36 of the supporting information, there are two instances where “flack” was written instead of “flask”.

=> We appreciate this suggestion. These typos were corrected in the revised version.

For Reviewer 2:

- (1) The manuscript by Chen et al. details the organization of a series of side-chain mesogenic polymers on Teflon sheets. The authors consider a broad combination of mesogen sequences as well as backbone chemistries, 14 samples in total, and assess the orientation of the columnar structures formed by the polymer chains (bottlebrushes) when hot pressed between Teflon sheets. The authors find that in all but two cases that homeotropic anchoring of the system is recovered. The two cases which do not show homeotropic anchoring form present $P6mm$ symmetry, whereas the homeotropic cases are of different symmetries. These two exceptions differ principally by the presence of an ether bond, rather than an ester bond, connecting the alkyl tail to the aromatic core of the terminal mesogen in the side chain. On this basis the authors assert that homeotropic anchoring is reliant on the presence of sufficiently strong inter-columnar dipole interaction which lead to the formation of a 2D rectangular lattice/tight bilateral connections, resulting in large sheet like assemblies that prefer to lie flat on a substrate.

=> We highly appreciate these encouraging remarks.

- (2) The data reported here represents something of a tour de force in terms of experimental effort. They represent the results of a systematic series of high quality experimental investigations. However, the conclusions drawn by the authors seem rather premature - the connection between dipole interactions and surface anchoring is tenuous: while one may accept that strong dipole-dipole interactions lead to large correlation lengths (i.e. "...large sheet like assembly..."), this is a different matter than accounting for the orientation of such an assembly at a given surface. Further, there are gaps/omissions in the data that preclude arrival at a firm conclusion regarding the origin of the differences in the anchoring condition:

=> We are afraid if our original manuscript could not convey our mechanistic claim correctly to this reviewer. Considering also the related comment raised by Reviewer 1, we added to the revised manuscript more elaborate mechanistic descriptions for the homeotropic orientation of bottlebrush polymers (page 9, lines 10–19).

=> In our previous paper (*Science*, **2010**, 330, 808), we reported that the monomer for PMA^{AAA} , when hot-pressed with Teflon sheets, aligns along the grooves of the Teflon sheets. We also reported that PMA^{AAA} in its 2D rectangular lattice intrinsically orients its mesogenic side chains mainly along the b -axis. Consequently, upon hot-pressing, polymer domains of PMA^{AAA} are unidirectionally oriented in such a way that the ab planes of their individual 2D rectangular lattices align parallel to the surface of the Teflon sheets. Hence, their c -axes align homeotropically with respect to the Teflon sheet surface. This represents how the homeotropic orientation of PMA^{AAA} is nucleated.

- => The important message of the present work is that the same orientation mechanism is applicable to other newly synthesized bottlebrush polymers (Fig. 4 and Supplementary Fig. 16) as well as their monomers (Supplementary Fig. 19).
- (3) The role of the pressing temperature/rate and of the surface grooves of the Teflon substrate have not been considered. For the purposes of investigating what molecular aspects contribute to planar vs homeotropic anchoring, the presence of potentially strong shear flows as well as topographical variation of the substrate are non-trivial confounding factors. We have no way of knowing whether the observed alignment is in fact one driven by shear flow and/or topography, rather than by some other effects. As pointed out above, a priori one does not have a reason to expect that dipole interactions would affect anchoring conditions, more so than say just correlation length, so one is left with more questions than answers in this respect.
- => In regard to the role of the surface grooves of the Teflon sheet, please see our answers to comment (2) raised by Reviewer 2.
- => Rod-shaped molecules or rod-shaped liquid crystalline mesogens are known to orient along micro-grooves of the rubbed polyimide films (*Chem. Mater.* **2003**, 15, 3105 [ref. 43]) or Teflon sheets (*Langmuir* **2001**, 17, 2192 [ref. 44]). Hence, it is very reasonable that the mesogens of the side chains of bottlebrush polymers prefer to be aligned along the grooves of Teflon sheets during hot-pressing. The revised version now includes these papers as references.
- => As described in our original manuscript (page 13, lines 2–8), we hot-pressed at 8 MPa all samples overnight at temperatures lower by 5 °C than the phase transition temperatures of individual bottlebrush polymers from their isotropic melt to ordered phase. This is as a result of the optimization for a better reproducibility. We did not care the pressing rate, considering that the hot-pressing is done for a rather long time (overnight), where the system is supposedly equilibrated.
- => We do not consider that the vertical hot-pressing, we employed for our experiments, can give rise to a shear flow for the bottlebrush polymers to stand up. Nevertheless, we newly conducted through-view 2D SAXS imaging of thermally annealed **PMA^{BBB}** and **PMA^{TTT}** films on a Teflon sheet after drop-casting without pressing. The obtained images did not show any 2D SAXS feature representing the homeotropic orientation of **PMA^{BBB}** and **PMA^{TTT}** (Supplementary Fig. 20) although the films were slightly birefringent along the surface grooves of the Teflon sheet. So, it is likely that the surface nucleation for the homeotropic polymer orientation may occur without pressing, but the pressing would assist the efficient propagation of the surface nucleation toward the interior of the film sample. These observations were added to the experimental section of the revised manuscript (page 13, lines 9–15).

(4) There is no cross-sectional scattering or TEM provided. The 4-fold symmetry in the POM could easily result from near-surface effects that may or may not be uniformly distributed across the systems.

=> We additionally obtained edge-view (cross-sectional) SAXS images of all the hot-pressed samples and added them to Supplementary Fig. 17 (new). As expected, the SAXS images of the bottlebrush polymers forming a rectangular lattice displayed only the (020) spots in the equatorial directions, while no diffraction spots were observed for the polymers forming a hexagonal lattice.

=> The revised manuscript now includes the following sentence; “Consistent with the results of through-view 2D SAXS imaging, only (020) spots were observed in the equatorial direction of the edge-view 2D SAXS image (Supplementary Fig. 17a).” (page 7, lines 3–5).

(5) The dipole moment is not quantified (e.g., by AFM measurements, or molecular modeling).

=> We estimated dipole moments of the mesogens using the density functional theory (DFT) method at the B3LYP/6-31G(d) level using a Gaussian 03 package (Fig. 2a).

(6) Beyond these issues, the paper as constructed does not do a good job of communicating the result presented. For example, the discussion on p. 11 regarding the pi-plane area of the mesogens was very difficult to follow. No consideration is given to the issue of molecular weight, not of flow alignment/other effects, as mentioned above.

=> As for the discussion on p. 11, what we would like to emphasize is a positive correlation between the total π -plane surface area of all the mesogen units in individual side chains and the thickness of the homeotropically ordered domains. As the total π -plane area of the side chain increases, the thickness of the homeotropically oriented domain from the Teflon sheet surface increases.

=> We utilized ATRP for obtaining low-PDI bottlebrush polymers but did not succeed in polymerizing the corresponding monomers. Because anionic polymerization is hopeless, we finally employed free-radical polymerization, affording polymers with non-uniform molecular weights. This does not allow us to discuss the effect of polymer molecular weight on the orientation behavior.

(4) The arguments in the introduction regarding the need for homeotropic anchoring of polymers at surfaces are not well formulated - in reality the desire for homeotropic anchoring is generally in self-assembled systems where one seeks vertical orientation of self-assembled structures, rather than individual polymer chains. Few readers will resonate with the idea that engineering homeotropic anchoring of polymer chains is a topic of intense concern.

=> We respectfully disagree with this reviewer’s comment. For example, Ma *et al.* (*J. Am. Chem. Soc.* 2013, **135**, 9644 [ref. 28]) reported that a homeotropically oriented poly(3-butylthiophene)

film exhibited more than 30-fold enhancement of the hole mobility compared to the pristine film. Because such semiconducting properties of polymeric materials are primarily determined by molecular orientations as well as molecular structures of polymers, it is clear that homeotropic orientation of individual polymers also accepts wide research interests in materials science, as highly evaluated by other two reviewers.

(5) Further, while the introduction suggests that a general strategy has been formulated for controlling polymer chain organization, in reality the work here has uncovered differences in the organization of columnar superstructures formed by large mesogenic side chains in a liquid crystalline bottlebrush system, when hot-pressed between grooved Teflon sheets, i.e. not a general strategy/phenomenon, but a highly specific scenario, which as discussed above, is not well rationalized from a fundamental perspective.

=> Based on the systematic study on the orientation behaviors of 14 different bottlebrush polymers, we think that we established a guiding principle for homeotropically orienting such polymers using the surface grooves of the Teflon sheet. Nevertheless, considering the criticism of this reviewer, we replaced the phrase “is quite universal for” with “may address” in the following sentence “Its underlying strategy and principle, unveiled for the first time in the present systematic study, is quite universal for a long-term question of how one can orient polymers homeotropically rather than horizontally”. Please note that, both in the abstract and introduction, we do not describe that we formulated a general strategy for controlling polymer chain organization.

For Reviewer 3:

- (1) The authors present an efficient method to orient polymers on substrates homeotropically. In 2010, they found that a bottlebrush polymer bearing three azobenzene units in their side chains could 'stand' on the substrate (*Science* **2010**, 330, 808). After a systematic study described in this manuscript, they reached a conclusion that the interaction between local dipoles in the side chains forces 'pitches' from neighboring cylindrical polymers to form a large 2D sheet-like assembly, and then contributed to the homeotropic alignment of polymer backbones. All the experiments are well designed and performed and this paper deserves an urgent publication in Nature Communication.

=> We highly appreciate these encouraging remarks.

- (2) Originality and interest: Novel
Data & methodology: Well-designed strategy and high quality of data presented in scientific way
quality of presentation
Appropriate use of statistics and treatment of uncertainties: Yes
Conclusions: Reliable
Suggested improvements: No revision
References: appropriate credit to previous work? yes

=> We highly appreciate these encouraging remarks.

Reviewers' Comments:

Reviewer #1 (Remarks to the Author)

The authors demonstrate through characterization large library of bottlebrush polymers that a highly robust mechanism for inducing homeotropic alignment of oblate cylindrical polymers has been elucidated. Both the synthesis and characterization aspects of the manuscript represent Herculean efforts. Oblate cylindrical polymers are shown to spontaneously orient on Teflon sheets with the cylindrical axis perpendicular to the surface (i.e., homeotropic alignment) when three polar mesogenic groups are incorporated in the side chains of bottlebrush polymers. Several permutations of the mesogen type and sequence of mesogens in the side chain as well as examples with different main chain polymer backbone demonstrate that the strategy has exceptionally wide scope. The fidelity with which the authors obtain homeotropic ordering among a series of analogous bottlebrush polymers should garner significant interest for both its fundamental and applied values.

The manuscript clearly shows that molecular design principles relate to macroscopic organization of polymers in homeotropically aligned domains. The critical number of polar mesogens required to obtain oblate rather than circular cylindrical polymers is established from X-ray characterization of a large number of bottlebrush polymers. Each of the six polymers that deviate from the design criterion show no mesophase or form circular cylindrical polymers organized in a columnar hexagonal lattice. All twelve bottlebrush polymers that fit the design criterion exhibit rectangular columnar mesophases made up from oblate cylindrical polymers. Each of these samples aligns homeotropically after melt pressing, and in all cases the same lattice direction aligns with the direction of the grooves in the Teflon substrate. A combination of three different techniques (polarized optical microscopy, polarized IR spectroscopy, and polarized fluorescence spectroscopy) further reveals that the mesogens in the side chains orient along the groove direction in the Teflon film, which is coincident with a particular lattice direction in the mesophase. Coincidence of the mesogen orientation with the groove direction is the key insight to how homeotropic alignment is nucleated. It should be further noted that the library of polymer that homeotropically align upon hot pressing includes two different rectangular columnar lattices, because this implies more generality to the strategy. Because the hot press processing method used to fabricate the homeotropically aligned polymers yields free-standing films, the strategy is not limited to a narrow set of applications.

With respect to revisions made by the authors, I am satisfied with the changes. The authors have clarified their ideas about the mechanism by which the ordering occurs during hot pressing between grooved Teflon sheets. The claims described in the preceding paragraph are still valid in the absence of analogs requested (PMAB'BB and PMAB'B'B). That the parameters of the hot press method are not varied to achieve homeotropic alignment of the 14 bottlebrush polymers speaks to how robust is the molecular design principles elucidated in this manuscript. Experiments testing these variables or discussion of these parameters seems beyond the scope of this manuscript. It is nice to see that the authors have added an estimate of the dipole moment for each mesogen in Figure 2a.

What causes the mesogens to align along the direction of the grooves in the Teflon film? This seems to be the key uncertainty in the manuscript. The revisions made by the authors help to clarify their explanation for nucleation. The authors propose that the net cross-sectional dipole of the bottlebrush polymer orients along the groove direction in the Teflon films, and this interaction serves to nucleate orientation of the bottlebrush polymers. Planar orientation of the bottlebrush polymers would preclude interaction of the net cross-sectional dipole with the grooves, while homeotropic alignment allows for the interaction. The net dipole arises from the anisotropic arrangement of mesogens in the oblate cylindrical polymers, and is therefore absent in circularly

cylindrical polymers. Strong interactions between the oblate cylinders appear to help ensure the formation of large monodomains of the aligned cylindrical polymers.

The discussion of the mechanism of homeotropic alignment is still hard to follow. In particular, the authors convolute formation of the monodomain with nucleation of homeotropic orientation. It is not clear that nucleation of homeotropic alignment requires large sheet-like assemblies that result from strong dipole-dipole interactions between mesogens on adjacent cylindrical polymers. Much of the manuscript talks about evidence for the latter, and very little discussion is devoted to the interaction that is needed to nucleate homeotropic alignment. As mentioned above, I think the authors have sufficient and convincing evidence for both nucleation and growth of large monodomains. The presentation of this information could be improved.

Reviewer #2 (Remarks to the Author)

The authors have made considerable revisions to their manuscript and have done good work in responding to the reviews. In particular the DFT and edge-on SAXS data help to clarify the assertions in the manuscript regarding the structure of the systems here. There are still open questions however. The responses regarding the role of the Teflon surface and any shear/topographical effects imparted by pressing against Teflon have not suitably resolved the issue raised in the prior round of review. This is all the more concerning as the manuscript advances an argument for a strategy to control homeotropic anchoring in general (due to dipole effects), rather than an argument about Teflon surfaces leading to homeotropic anchoring. Given the arguments advanced by the authors, a robust confirmation of the ideas presented here would involve demonstration of homeotropic order on multiple different surfaces - even if the resulting grain sizes are not large, the orientation of the c-axis should be uniform. The authors seem to discount the possibility, for example, that the orientation is due to particular combinations of shear rate/temperature and substrate topography during melt pressing of the films. In the absence of a more comprehensive exploration that links dipole interactions to orientational control *independent* of the substrate/pressing, the data presented do not substantiate the claims made in the manuscript as currently constructed. It is clear that the authors have uncovered a relationship between molecular structure and orientation for samples subjected to a certain processing scheme, but successfully making the broader claims that they've made here requires more validation.

The response to the question itself is confusing:

"Nevertheless, we newly conducted through-view 2D SAXS imaging of thermally annealed PMABBB and PMATTT films on a Teflon sheet after drop-casting without pressing. The obtained images did not show any 2D SAXS feature representing the homeotropic orientation of PMABBB and PMATTT (Supplementary Fig. 20) although the films were slightly birefringent along the surface grooves of the Teflon sheet. So, it is likely that the surface nucleation for the homeotropic polymer orientation may occur without pressing, but the pressing would assist the efficient propagation of the surface nucleation toward the interior of the film sample."

From the above narrative I infer that drop casting did not produce homeotropic ordering, but in the last sentence the authors assert that nucleation for homeotropic ordering may occur without pressing (i.e. in drop casting). These statements are contradictory.

The authors are correct that in some cases there is a desire for homeotropic ordering of polymer chains themselves -the semiconducting polymer example is a good one. However in most cases, the desire is for control of orientation of some structure that is formed by polymers. Eg. For the membrane example discussed in the introduction, at least for block copolymer based membranes, the chains are actually parallel to film plane, which results in the desired vertical orientation of nanopores. I urge the authors to streamline and clarify their narrative regarding controlling orientation of chains vs structures, and to properly represent what has been done relative to what

the broader challenge is - the statement at the end of the paragraph at the top of p4 that "This guiding principle provides a clear answer to the long-standing issue in polymer science." suggests to the reader that a general principle has been arrived at, but this is not the case.

For these reasons, publication in Nature Communication as currently constructed is not recommended.

[2] Point-to-Point Answers to Reviewers' Comments

For Reviewer 1:

- (1) The authors demonstrate through characterization large library of bottlebrush polymers that a highly robust mechanism for inducing homeotropic alignment of oblate cylindrical polymers has been elucidated. Both the synthesis and characterization aspects of the manuscript represent Herculean efforts. Oblate cylindrical polymers are shown to spontaneously orient on Teflon sheets with the cylindrical axis perpendicular to the surface (i.e., homeotropic alignment) when three polar mesogenic groups are incorporated in the side chains of bottlebrush polymers. Several permutations of the mesogen type and sequence of mesogens in the side chain as well as examples with different main chain polymer backbone demonstrate that the strategy has exceptionally wide scope. The fidelity with which the authors obtain homeotropic ordering among a series of analogous bottlebrush polymers should garner significant interest for both its fundamental and applied values.

=> We highly appreciate these encouraging remarks.

- (2) The manuscript clearly shows that molecular design principles relate to macroscopic organization of polymers in homeotropically-aligned domains. The critical number of polar mesogens required to obtain oblate rather than circular cylindrical polymers is established from X-ray characterization of a large number of bottlebrush polymers. Each of the six polymers that deviate from the design criterion show no mesophase or form circular cylindrical polymers organized in a columnar hexagonal lattice. All twelve bottlebrush polymers that fit the design criterion exhibit rectangular columnar mesophases made up from oblate cylindrical polymers. Each of these samples aligns homeotropically after melt pressing, and in all cases the same lattice direction aligns with the direction of the grooves in the Teflon substrate. A combination of three different techniques (polarized optical microscopy, polarized IR spectroscopy, and polarized fluorescence spectroscopy) further reveals that the mesogens in the side chains orient along the groove direction in the Teflon film, which is coincident with a particular lattice direction in the mesophase. Coincidence of the mesogen orientation with the groove direction is the key insight to how homeotropic alignment is nucleated. It should be further noted that the library of polymer that homeotropically align upon hot pressing includes two different rectangular columnar lattices, because this implies more generality to the strategy. Because the hot press processing method used to fabricate the homeotropically aligned polymers yields free-standing films, the strategy is not limited to a narrow set of applications.

=> We highly appreciate these encouraging remarks.

- (3) With respect to revisions made by the authors, I am satisfied with the changes. The authors have clarified their ideas about the mechanism by which the ordering occurs during hot pressing between grooved Teflon sheets. The claims described in the preceding paragraph are still valid in the absence of analogs requested ($\text{PMA}^{\text{B'BB}}$ and $\text{PMA}^{\text{B'B'B}}$). That the parameters of the hot press method are not varied to achieve homeotropic alignment of the 14 bottlebrush polymers speaks to how robust is the molecular design principles elucidated in this manuscript. **Experiments testing these variables or discussion of these parameters seem beyond the scope of this manuscript.** It is nice to see that the authors have added an estimate of the dipole moment for each mesogen in Figure 2a.

=> We highly appreciate these encouraging remarks. We really think that our molecular design principle is robust for achieving the homeotropic alignment.

- (4) What causes the mesogens to align along the direction of the grooves in the Teflon film? This seems to be the key uncertainty in the manuscript. The revisions made by the authors help to clarify their explanation for nucleation. The authors propose that the net cross-sectional dipole of the bottlebrush polymer orients along the groove direction in the Teflon films, and this interaction serves to nucleate orientation of the bottlebrush polymers. Planar orientation of the bottlebrush polymers would preclude interaction of the net cross-sectional dipole with the grooves, while homeotropic alignment allows for the interaction. The net dipole arises from the anisotropic arrangement of mesogens in the oblate cylindrical polymers, and is therefore absent in circularly cylindrical polymers. Strong interactions between the oblate cylinders appear to help ensure the formation of large monodomains of the aligned cylindrical polymers.

The discussion of the mechanism of homeotropic alignment is still hard to follow. In particular, the authors convolute formation of the monodomain with nucleation of homeotropic orientation. It is not clear that nucleation of homeotropic alignment requires large sheet-like assemblies that result from strong dipole-dipole interactions between mesogens on adjacent cylindrical polymers. Much of the manuscript talks about evidence for the latter, and very little discussion is devoted to the interaction that is needed to nucleate homeotropic alignment. As mentioned above, I think the authors have sufficient and convincing evidence for both nucleation and growth of large monodomains. The presentation of this information could be improved.

=> We totally agree with and appreciate this constructive comment.

=> According to the comment raised by reviewer 1, we revised a part of the manuscript featuring the mechanism of how the homeotropic orientation of the bottlebrush polymer takes place. In particular, we removed one of our claims that the large sheet-like domain may contribute to the homeotropic orientation of the polymers.

As a consequence, the revised manuscript now contains following two important mechanistic elements:

- [1] As discussed for the monomer of **PMA^{AAA}** in our previous manuscript, those of **PMA^{BBB}** and **PMA^{TTT}**, when hot-pressed, orient their mesogenic units along the surface grooves of Teflon sheets. Also noteworthy, the corresponding bottlebrush polymers orient their mesogenic side chains mainly along the *b*-axis of the 2D rectangular lattice. Consequently, polymer domains are oriented unidirectionally in such a way that the *ab* planes of their individual 2D rectangular lattices align parallel to the surface of the Teflon sheets. Hence, their *c*-axes align homeotropically with respect to the surface of the Teflon sheets.
- [2] With a physical assistance of the surface grooves on the Teflon sheets, nucleation for homeotropic ordering can be induced and propagate efficiently upon hot-pressing toward the interior of the film, wherein consistent polymer molecules align homeotropically.

For Reviewer 2:

- (1) The authors have made considerable revisions to their manuscript and have done good work in responding to the reviews. In particular, the DFT and edge-on SAXS data help to clarify the assertions in the manuscript regarding the structure of the systems here.

=> We highly appreciate these encouraging remarks.

- (2) There are still open questions however. The responses regarding the role of the Teflon surface and any shear/topographical effects imparted by pressing against Teflon have not suitably resolved the issue raised in the prior round of review. This is all the more concerning as the manuscript advances an argument for a strategy to control homeotropic anchoring in general (due to dipole effects), rather than an argument about Teflon surfaces leading to homeotropic anchoring. Given the arguments advanced by the authors, a robust confirmation of the ideas presented here would involve demonstration of homeotropic order on multiple different surfaces - even if the resulting grain sizes are not large, the orientation of the *c*-axis should be uniform.

=> According to the comment raised by reviewer 1, we revised a part of the manuscript featuring the mechanism of how the homeotropic orientation of the bottlebrush polymer takes place. In particular, we removed one of our claims that the large sheet-like domain may contribute to the homeotropic orientation of the polymers.

As a consequence, the revised manuscript now contains following two important mechanistic elements:

[1] As discussed for the monomer of **PMA^{AAA}** in our previous manuscript, those of **PMA^{BBB}** and **PMA^{TTT}**, when hot-pressed, orient their mesogenic units along the surface grooves of Teflon sheets. Also noteworthy, the corresponding bottlebrush polymers orient their mesogenic side chains mainly along the *b*-axis of the 2D rectangular lattice. Consequently, polymer domains are oriented unidirectionally in such a way that the *ab* planes of their individual 2D rectangular lattices align parallel to the surface of the Teflon sheets. Hence, their *c*-axes align homeotropically with respect to the surface of the Teflon sheets.

[2] With a physical assistance of the surface grooves on the Teflon sheets, nucleation for homeotropic ordering can be induced and propagate efficiently upon hot-pressing toward the interior of the film, wherein consistent polymer molecules align homeotropically.

- (3) The authors seem to discount the possibility, for example, that the orientation is due to particular combinations of shear rate/temperature and substrate topography during melt pressing of the films. In the absence of a more comprehensive exploration that links dipole interactions to

orientational control “independent” of the substrate/pressing, the data presented do not substantiate the claims made in the manuscript as currently constructed. It is clear that the authors have uncovered a relationship between molecular structure and orientation for samples subjected to a certain processing scheme, but successfully making the broader claims that they’ve made here requires more validation.

=> In regard to the type of substrate, our method is not universal but requires Teflon sheets. So long as polymer samples are sandwiched by Teflon sheets such that their surface grooves are oriented parallel to one another, constituent bottlebrush polymers align homeotropically using a very traditional hot-press machine that hardly controls the pressing rate (see right).

=> As described for Comment (4) raised by reviewer 1, we revised the mechanistic discussion for the homeotropic orientation of bottlebrush polymers.

(4) The response to the question itself is confusing:

“Nevertheless, we newly conducted through-view 2D SAXS imaging of thermally annealed **PMA^{BBB}** and **PMA^{TTT}** films on a Teflon sheet after drop-casting without pressing. The obtained images did not show any 2D SAXS feature representing the homeotropic orientation of **PMA^{BBB}** and **PMA^{TTT}** (Supplementary Fig. 20) although the films were slightly birefringent along the surface grooves of the Teflon sheet. So, it is likely that the surface nucleation for the homeotropic polymer orientation may occur without pressing, but the pressing would assist the efficient propagation of the surface nucleation toward the interior of the film sample.”

From the above narrative, I infer that drop casting did not produce homeotropic ordering, but in the last sentence the authors assert that nucleation for homeotropic ordering may occur without pressing (i.e. in drop casting). These statements are contradictory.

=> Again, in regard to the type of substrate, our method is not universal but requires Teflon sheets.

=> We wonder if this reviewer may confuse “homeotropic ordering” and “nucleation for homeotropic ordering” in our statements. Our statement means that “nucleation for homeotropic ordering” can be induced by the surface grooves of the Teflon sheet and does not require “hot-pressing”, but “hot-pressing” would certainly assist the efficient propagation of this “surface nucleation” toward the interior of the film sample to achieve “homeotropic ordering of polymers”. We do not think that these two statements are contradictory to one another.

- (5) The authors are correct that in some cases there is a desire for homeotropic ordering of polymer chains themselves the semiconducting polymer example is a good one. However, in most cases, the desire is for control of orientation of some structure that is formed by polymers (eg. For the membrane example discussed in the introduction, at least for block copolymer based membranes, the chains are actually parallel to film plane, which results in the desired vertical orientation of nanopores) I urge the authors to streamline and clarify their narrative regarding controlling orientation of chains vs structures, and to properly represent what has been done relative to what the broader challenge is - the statement at the end of the paragraph at the top of p4 that “This guiding principle provides a clear answer to the long-standing issue in polymer science.” suggests to the reader that a general principle has been arrived at, but this is not the case.
- => Once again, we have no idea of informing that our method that requires hot-pressing with Teflon sheets is universal. The purpose of this manuscript is to report why and how bottlebrush polymers of particular structures, when sandwiched by Teflon sheets, align homeotropically upon hot-pressing. As clearly written in the introductory part, the present work simply follows our previous work and addresses a very interesting and unsolved question on the orientation mechanism.
- => Since the manuscript is not a review article for orientation of polymers, the above request raised by this reviewer seems to be beyond the scope of this manuscript.
- => We removed “guiding” from the following sentence (p. 4, lines 2–3); This guiding principle provides a clear answer to the long-standing issue in polymer science.

Reviewers' Comments:

Reviewer #2 (Remarks to the Author)

I regret to say, but it is my finding that this manuscript is not suitable for publication in Nature Communications. A careful review of the comments by the referees reveals non-trivial concerns regarding the representation of the results, and the revisions in this regard have been somewhat cursory and overall insufficient to correct the issues identified. Speculation regarding the role of the surface grooves is presented as established fact, while the manuscript is written in a way that suggests that a universal solution/approach has been arrived at, when it is simply not the case (e.g. in the introduction: "...This principle provides a clear answer to one of the long-standing issues in polymer science...").

Finally, the manner in which the data is discussed is confusing - the issues regarding rectangular vs hexagonal packing, and the issue of homeotropic order have been conflated to some extent. In reality, homeotropic backbone ordering in a liquid crystalline polymer is not that unusual as the anchoring condition/flow behavior of the LC structure can dictate the behavior of the polymer backbone. I renew my concern that the authors here treat the question of backbone orientation separately from the LC anchoring behavior. At the end of the day, the manuscript communicates a plethora of high quality and interesting data, but there is a persistent disconnect between the data and its interpretation, and it is in the interpretation and presentation of results that the impact should materialize, but here the manuscript is lacking.

Point-to-Point Answers to Reviewer #2

Before starting our point-to-point answers to his/her criticisms and comments, we would like to inform to this referee about what we have newly done additionally. As summarized in Fig. C1, we carried out one systematic study on how the pressing time affects the extent of homeotropic ordering of **PMA^{TTT}**. What we found is that the homeotropic ordering hardly takes place shortly after the hot-pressing is made and requires the sample to be kept pressed for several hours to be accomplished. This may indicate that the homeotropic ordering is not as the result of a kinetic process but a thermodynamic process nucleated by the unidirectionally oriented grooves on the Teflon sheets. As for the role of the surface grooves, we have reported in our previous paper in *Science* that the *b*-axis of the 2D rectangular lattice (direction of the oriented mesogenic units), formed by hot-pressing, always aligns parallel to the direction of the grooves, despite the fact that the sample is stretched into a thin film concentrically. As also reported in the same paper, when the upper and lower Teflon sheets are orthogonal in terms of their groove directions, the resultant hot-pressed film composed of an azobenzene bottlebrush polymer does not undergo any photomechanical bending because of an orientational confusion between the front and backsides of the film. So, the surface grooves indeed play a dominant role, although how the molecule interacts with such surface grooves is a type of subject, which is highly challenging even now.

Reviewer #2 insisted that we should investigate how the pressing (shear) rate affects the extent of ordering. Such a hot-press machine with a precise controller for the pressing rate is not available because of no industrial demand. So, as an alternative, we decided to newly perform the above systematic study summarized in Fig. C1. We do not say that the shear, generated concomitantly upon pressing, is unimportant but would like to claim that **the grooves on the Teflon sheets play a dominant role, where bottlebrush polymers that assemble into a 2D rectangular lattice slowly align homeotropically (the *b*-axis of the 2D rectangular lattice aligns along the direction of the grooves) while the samples are kept pressed.** No homeotropic ordering takes place shortly after hot-pressing. Some minor structuring takes place simply upon drop casting the sample on a Teflon sheet, but this minor structure does not develop without hot-pressing.

Interfacial phenomena are exciting but still challenging even now. At any rate, we have no doubt that, with such newly obtained data, all three referees including referee #2 would be happy to recommend publication of our work in *Nature Commun.* Note that, due to the idle period of the Japanese synchrotron radiation facility SPring-8, we have at present no choice but to use a SAXS machine in our laboratory. Hence, the quality of the SAXS data we obtained were lower than those of other SAXS data in the manuscript taken at SPring-8. Nevertheless, the data are qualified enough to discuss about how the orientation event proceeds with the hot-pressing time.

a 2D SAXS Images

b 1D SAXS Patterns

c Angular Dependency of the Peak Intensity (d_{020})

Figure C1. (a) Through-view 2D SAXS images of thin films of PMA^{TTT} hot-pressed for 2, 5, 30, 60, 180 and 480 min. Green arrows represent the directions of the grooves on the Teflon sheets. 1D SAXS patterns (b) and angular dependency (c) of the peak intensity of the diffraction from the (020) plane converted from the SAXS images of the films.

- (1) I regret to say, but it is my finding that this manuscript is not suitable for publication in Nature Communications. A careful review of the comments by the referees reveals non-trivial concerns regarding the representation of the results, and the revisions in this regard have been somewhat cursory and overall insufficient to correct the issues identified.
- => In our understanding, other two reviewers are apparently positive for the acceptance of our manuscript. We do not understand what non-trivial concerns they have revealed.
- => No hot-press machines capable of tuning the pressing speed are available. Alternatively, we newly performed a systematic study on a bottlebrush polymer **PMA^{TTT}** to clarify how the hot-pressing time affects the degree of homeotropic order (Fig. C1). What we found is that the homeotropic ordering hardly takes place shortly after the hot-pressing is made and requires the sample to be kept pressed for several hours to be accomplished.
- (2) Speculation regarding the role of the surface grooves is presented as established fact, while the manuscript is written in a way that suggests that a universal solution/approach has been arrived at, when it is simply not the case (*e.g.*, in the introduction: "...This principle provides a clear answer to one of the long-standing issues in polymer science...").
- => The role of surface grooves was described clearly in our previous paper (*Science* 2010, **330**, 808). For quick understanding of reviewer #2, we provided at the end of this cover letter two lists, one of which features our experimental observations made in the present work and the other summarizes the achievements reported in our previous *Science* paper. Importantly, the *b*-axis of the 2D rectangular lattice aligns parallel to the surface grooves on the Teflon sheet. As also reported in the same paper, when the upper and lower Teflon sheets are orthogonal to one another in terms of their groove directions, the resultant hot-pressed film composed of azobenzene-containing PMA^{AAA} does not undergo any photomechanical bending because of an orientational confusion between the front and backsides of the film.
- => As stated clearly in the first and second rounds of review, we no longer use the term "universal" and "guiding principle" (we strongly hope this reviewer to go through all the details of the submitted documents carefully). Nevertheless, we decided to revise further the indicated sentence as "The first systematic study presented herein may provide one of promising molecular design strategies for polymers that align homeotropically on substrate surfaces".
- (3) Finally, the manner in which the data is discussed is confusing - the issues regarding rectangular vs hexagonal packing, and the issue of homeotropic order have been conflated to some extent.
- => Please go through our original manuscript once again (page 7, line 23–page 8, line 10). We clearly stated that the bottlebrush polymers with a 2D rectangular geometry are aligned homeotropically upon hot-pressing, whereas those with a 2D hexagonal geometry are randomly oriented even after hot-pressing.

(4) In reality, homeotropic backbone ordering in a liquid crystalline polymer is not that unusual as the anchoring condition/flow behavior of the LC structure can dictate the behavior of the polymer backbone. I renew my concern that the authors here treat the question of backbone orientation separately from the LC anchoring behavior.

=> This message is wrong. As suggested by reviewer #1 (supposedly an expert for LC polymers), there have been reported only two LC polymers, both of which coincidentally carry a dendritic side chain and spontaneously orient their backbone homeotropically on a substrate (*Nature* 2002, **419**, 382, *Macromolecules* 2015, **48**, 2849). Before our present work, no systematic study has been done because of an extremely small number of homeotropically orientable polymers. No prediction about the backbone orientation is possible from the polymer molecular structure even though it has LC properties or mesogens.

(5) At the end of the day, the manuscript communicates a plethora of high quality and interesting data, but there is a persistent disconnect between the data and its interpretation, and it is in the interpretation and presentation of results that the impact should materialize, but here the manuscript is lacking.

=> Our understanding is that the guideline of nature and its sister journals for reviewers does not allow such highly ambiguous criticisms. This may also hold true in the criticism (3).

For quick understanding of reviewer #2, we provided the following lists, one features our experimental observations in the present work and the other summarizes the achievements reported in our previous Science paper.

List of our experimental observations in the present work:

- (1) Twelve bottlebrush polymers that self-assemble into a 2D rectangular lattice, upon hot-pressing in Teflon sheets, align homeotropically, while two bottlebrush polymers that self-assemble into a 2D hexagonal lattice, do not align homeotropically (though-view and edge-view 2D SAXS data, FT-IR, POM, fluorescent anisotropy).
- (2) In each successful case, the *b*-axis of the 2D rectangular lattice aligns along the surface grooves on a Teflon sheet.
- (3) In relation to (2), even the monomers of **PMA^{BBB}** and **PMA^{TTT}**, after hot-pressing, orient their mesogenic units along the surface grooves on Teflon sheets.
- (4) Shortly after the hot pressing, the homeotropic order of **PMA^{TTT}** hardly emerges, but requires the sample to be kept hot-pressed for several hours to be accomplished. ---- new data (Figure C1 in this document).

- (5) Some birefringent minor structuring, which takes place simply upon drop-casting **PMA^{BBB}** and **PMA^{TTT}** on a Teflon sheet, emerges along the surface grooves on a Teflon sheet, but this minor structure does not develop without hot-pressing.

List of what we highlighted in *Science* 2010, 330, 808:

- [1] Homeotropic orientation of a bottlebrush polymer over a macroscopic length scale was realized for azobenzene-containing **PMA^{AAA}** by hot-pressing with Teflon sheets.
=> On both sides of the whole processed film, the polymer main chain aligns homeotropically to the film plane, while the grafted side chains align horizontally along the surface grooves on the Teflon sheets.
- [2] Directional control of the oriented grafted side chains on each side of the processed film:
=> Orientation direction of the grafted side chains can be controlled freely, on each side of the processed film, depending on the surface grooves of the Teflon sheet for hot pressing.
- [3] The processed film with such a bimorph configuration is referred to as a 'soft actuator' responsive to light.
=> Photomechanical bending takes place when the directions of surface grooves on two Teflon sheets, used for the hot-pressing, are parallel to one another. In contrast, no photomechanical bending takes place when they are orthogonal to one another.
- [4] In regard to the photomechanical bending, effects of residual strains generated on individual sides of the hot-pressed films in regards to the surface groove directions were discussed.

Reviewers' Comments:

Reviewer #3 (Remarks to the Author)

Ordering of polymers is an important issue in soft matter science, not only due to its fundamental structural characterization, but also its effect on materials' mechanic behaviors, and thus in industrial applications. The authors demonstrate a molecular design principle for inducing homeotropic alignment of bottlebrush polymers. In order to explain their observations, they have provided a reliable mechanism which has been supported by their experimental results. This is a systematic research with a set of well-designed and carefully conducted experiments.

With respect to the comments made by reviewer #2, the main concern raised is that, during hot-pressing against Teflon, whether any shear/topographical effects will cause the alignment of the mesogenic units in polymer side chains along surface grooves. Although the reviewer #2's comment is appreciated, there are a number of publications regarding orientation of polymer chains on the elongated PTFE film surfaces with/without hot pressing, indicating the physical groove-alignment effect at the interfaces, which serves as nucleation to gradually develop towards inner film along the direction of film normal. For polymers with side chain liquid crystals as in the current case, the flexible backbones aligned homeotropically is truly a new observation. An alignment of mesogenic side chains induced by chemical/physical interactions between polymers and PTFE at the interface does not warrant the homeotropic orientation of the flexible backbone chains. There must be another mechanism to cause this homeotropic backbone alignment. The proposed mechanism of the dipole interactions of the mesogenic groups generated by their anisotropic arrangement (the rectangular lattice constructed by non-circular motives) along the orientational axis of elongated PTFE films is a reasonable one. As described in the method section, the nucleation for the rectangular lattice can be induced at the interface of oriented Teflon sheet and then, propagate efficiently to large monodomains during the hot-pressing (see Section 'Method'). This reviewer welcomes that the authors have provided time-dependence SAXS results in the latest cover letter to further strength their claim. Judging from Figure C1c, after 30 minutes hot-pressing (the free-standing film has been formed with a fixed shape), there is only minor amount of ordering structure in the resulting film induced by the oriented PTFE at the interface, indicating that a prolonged annealing time is needed (up to 8 hours) to grow the rectangular lattice structure in the entire film. As a result, the shearing force induced by the hot-pressing and physical groove structure at the PTFE film surface only stimulate the initial nucleation process in the ordering of the side chain liquid crystal mesogens.

Point-to-Point Answers to Reviewer #3

- (1) Ordering of polymers is an important issue in soft matter science, not only due to its fundamental structural characterization, but also its effect on materials' mechanic behaviors, and thus in industrial applications. The authors demonstrate a molecular design principle for inducing homeotropic alignment of bottlebrush polymers. In order to explain their observations, they have provided a reliable mechanism which has been supported by their experimental results. This is a systematic research with a set of well-designed and carefully conducted experiments.

=> We really appreciate this encouraging comments.

- (2) With respect to the comments made by reviewer #2, the main concern raised is that, during hot-pressing against Teflon, whether any shear/topographical effects will cause the alignment of the mesogenic units in polymer side chains along surface grooves. Although the reviewer #2's comment is appreciated, there are a number of publications regarding orientation of polymer chains on the elongated PTFE film surfaces with/without hot pressing, indicating the physical groove-alignment effect at the interfaces, which serves as nucleation to gradually develop towards inner film along the direction of film normal. For polymers with side chain liquid crystals as in the current case, the flexible backbones aligned homeotropically is truly a new observation. An alignment of mesogenic side chains induced by chemical/physical interactions between polymers and PTFE at the interface does not warrant the homeotropic orientation of the flexible backbone chains. There must be another mechanism to cause this homeotropic backbone alignment. The proposed mechanism of the dipole interactions of the mesogenic groups generated by their anisotropic arrangement (the rectangular lattice constructed by non-circular motives) along the orientational axis of elongated PTFE films is a reasonable one. As described in the method section, the nucleation for the rectangular lattice can be induced at the interface of oriented Teflon sheet and then, propagate efficiently to large monodomains during the hot-pressing (see Section 'Method'). This reviewer welcomes that the authors have provided time-dependence SAXS results in the latest cover letter to further strength their claim. Judging from Figure C1c, after 30 minutes hot-pressing (the free-standing film has been formed with a fixed shape), there is only minor amount of ordering structure in the resulting film induced by the oriented PTFE at the interface, indicating that a prolonged annealing time is needed (up to 8 hours) to grow the rectangular lattice structure in the entire film. As a result, the shearing force induced by the hot-pressing and physical groove structure at the PTFE film surface only stimulate the initial nucleation process in the ordering of the side chain liquid crystal mesogens.

=> Thank you for your kind support.